# Non-Digestible Oligosaccharides and Short Chain Fatty Acids as Therapeutic Targets against Enterotoxin-Producing Bacteria and Their Toxins

**DOI:** 10.3390/toxins13030175

**Published:** 2021-02-25

**Authors:** Mostafa Asadpoor, Georgia-Nefeli Ithakisiou, Paul A. J. Henricks, Roland Pieters, Gert Folkerts, Saskia Braber

**Affiliations:** 1Division of Pharmacology, Utrecht Institute for Pharmaceutical Sciences, Faculty of Science, Utrecht University, Universiteitsweg 99, 3584 CG Utrecht, The Netherlands; m.asadpoor@uu.nl (M.A.); g.ithakisiou@students.uu.nl (G.-N.I.); p.a.j.henricks@uu.nl (P.A.J.H.); g.folkerts@uu.nl (G.F.); 2Division of Medicinal Chemistry and Chemical Biology, Utrecht Institute for Pharmaceutical Sciences, Faculty of Science, Utrecht University, Universiteitsweg 99, 3584 CG Utrecht, The Netherlands; R.J.Pieters@uu.nl

**Keywords:** enterotoxins, enteropathogenic bacteria, oligosaccharides, short chain fatty acids

## Abstract

Enterotoxin-producing bacteria (EPB) have developed multiple mechanisms to disrupt gut homeostasis, and provoke various pathologies. A major part of bacterial cytotoxicity is attributed to the secretion of virulence factors, including enterotoxins. Depending on their structure and mode of action, enterotoxins intrude the intestinal epithelium causing long-term consequences such as hemorrhagic colitis. Multiple non-digestible oligosaccharides (NDOs), and short chain fatty acids (SCFA), as their metabolites produced by the gut microbiota, interact with enteropathogens and their toxins, which may result in the inhibition of the bacterial pathogenicity. NDOs characterized by diverse structural characteristics, block the pathogenicity of EPB either directly, by inhibiting bacterial adherence and growth, or biofilm formation or indirectly, by promoting gut microbiota. Apart from these abilities, NDOs and SCFA can interact with enterotoxins and reduce their cytotoxicity. These anti-virulent effects mostly rely on their ability to mimic the structure of toxin receptors and thus inhibiting toxin adherence to host cells. This review focuses on the strategies of EPB and related enterotoxins to impair host cell immunity, discusses the anti-pathogenic properties of NDOs and SCFA on EPB functions and provides insight into the potential use of NDOs and SCFA as effective agents to fight against enterotoxins.

## 1. Introduction

Currently, impairment of the gastrointestinal tract caused by the activity of bacterial enteropathogens is one of the biggest issues affecting human health and food safety [1]. Because of a growing concern on the relationship between toxigenic bacteria and intestinal associated diseases, research is required to define conditions and minimize the levels of their toxicity. Dietary carbohydrates, especially non-digestible oligosaccharides (NDOs) and short chain fatty acids (SCFA) as their metabolites produced by the gut microbiota, are known to reduce the toxic potential of bacterial enteropathogens in multiple stages of their pathogenicity [2]. Furthermore, the NDOs possess important physiological and physicochemical properties and serve as dietary fibers and prebiotics. Additionally, NDOs of various origins have been used extensively as immunostimulators, animal feed, agrochemicals, cosmetics and for drug delivery [3].

The human gut microbiota harbors a diverse community of commensal bacteria with a vast biosynthetic capacity. The role of the microbiota and its residents, is essential for the host, since it regulates multiple functions, including immune system development, nutrient processing and prevention of pathogen colonization [4]. The intestinal gut microbiota is directly exposed to the external environment, and therefore highly susceptible to pathogenic invasion and colonization [5]. Intestinal epithelial cells can be targeted by various pathogenic bacteria and consequently by their virulence factors, such as, toxins. More specifically, toxins secreted by bacteria that selectively interact with intestinal cells are called enterotoxins. Following different modes of actions, including pore formation, increase in permeability of the intestinal epithelium and alterations in cell homeostasis, enterotoxins can cause different gastrointestinal diseases such as pseudomembranous colitis [6]. Among the major pathogenic bacteria that secrete highly toxic proteins, enterotoxin-producing *Bacillus cereus*, *Clostridium difficile*, *Clostridium perfringens*, *Escherichia coli*, *Staphylococcus aureus* and *Vibrio cholerae* are the most prominent. Therefore, a clear understanding of key features of toxicogenic bacteria as well as their virulent products is required for the development and selection of optimal treatments.

To date, antibiotics are the most promising therapy for diseases related to enterotoxin-producing bacteria (EPB), however, exponential use and misuse of antibiotics have led to loss of their efficacy and antimicrobial resistance [7]. Several mechanisms of antibiotic-resistant pathogenic bacteria render these antimicrobials inactive and prolong their survival and pathogenicity by, for example, biofilm formation. Since many infections remain untreated, antibiotic resistance in bacterial pathogens is one of the great challenges in the developed and developing world with immense clinical and economic impacts [8]. Therefore, new strategies to resolve this escalating problem and diminish bacterial resistance-associated infections are urgently needed. Over recent decades, there is a growing interest in NDOs as anti-pathogenic agents, since NDOs do not only maintain gut homeostasis, but can also exert microbiota-independent effects on intestinal epithelial and immune cells with minimum side effects [9].

NDOs obtained from natural sources or manufactured via enzymatic or chemical synthesis, can get fermented by the beneficial bacteria to release metabolic substrates and energy [10]. Additionally, according to their key characteristics, such as, monosaccharide building blocks, degree of polymerization (DP), degree of acetylation (DA) and charge, they exhibit anti-pathogenic effects in multiple ways. Antimicrobial capabilities of NDOs are not limited against pathogenic bacteria, but also anti-virulent properties by blocking virulence factors, such as enterotoxins, have been described. Receptor mimicry mechanisms and stimulation or blocking of intracellular pathogenic mechanisms are some of the strategies that NDOs use to encounter such toxins. SCFA, as the end products of oligosaccharide fermentation induced by anaerobic intestinal microbiota, can induce similar anti-toxic effects, but through different mechanisms. Among diverse SCFA, acetate, propionate and butyrate have shown the most prominent anti- pathogenic effects [11].

This review aims to explore the current state of knowledge on the anti-virulence strategies of NDOs and SCFA against pathogenic bacteria and associated enterotoxins that target the intestinal epithelial layer. The review starts with describing structural characteristics and mode of action of the major virulent enterotoxins, including enterotoxins related to *B. cereus*, *C. difficile* and *C. perfringens*, cholera toxin (CT), heat-labile (LT) and heat-stable (ST) enterotoxins, Shiga toxins (Stxs) and *staphylococcal* enterotoxins (SEs). Thereafter, the main characteristics of the described NDOs are presented followed by a comprehensive overview of the anti-microbial functionalities of NDOs and SCFA against EPB and their enterotoxins. First, the anti-pathogenic properties of NDOs and SCFA against the EPB are specified “directly” by exerting anti-adhesive, anti-biofilm and anti-growth effects against EPB and “indirectly” by promoting the growth of beneficial bacteria that maintain gut homeostasis, mainly through SCFA production, which results in a reduction in final colonization and prefiltration of EPB. Second, the mechanisms of action (both direct and indirect) of NDOs and SCFA against each enterotoxin are discussed, which may open new avenues for NDOs and SCFA as effective agents to fight against enterotoxins.

## 2. Enterotoxin-Producing Bacteria and Related Enterotoxins

The pathogenicity of various bacteria on the human intestinal epithelium is associated with their ability to produce certain virulence factors, called enterotoxins. Enterotoxins tend to be produced by Gram-positive bacteria, however, some exceptions of Gram-negative bacteria, such as *E. coli* and *V. cholerae,* are also characterized as enterotoxin-producing bacteria. According to their structure and pathogenic characteristics, enterotoxins can cause different pathogenic effects, such as the disturbance of the cellular ionic balance due to membrane pore formation and overstimulation of the immune response. A detailed investigation of the characteristics and the cytotoxicity pathways of both enterotoxin-producing bacteria and enterotoxins can provide insights in potential therapeutic treatments.

### 2.1. B. cereus and Related Enterotoxins

#### 2.1.1. Epidemiology of *B. cereus* and Related Enterotoxins

*B. cereus*, a Gram-positive microorganism, is a versatile, spore-forming and facultative anaerobe abundantly found in nature, most commonly isolated from soil, growing plants, as well as in food production environments. *B. cereus* is a motile bacterium with a wide growth temperature range (8–55 °C). Its ability to form spores render it highly resistant to harsh environmental conditions, such as low pH values, drought or radiation [12].

The spread from bacterium habitats to foods and therefore to humans can cause two types of foodborne disease: an intoxication (emetic form) and a toxico-infection (diarrheal form). At the diarrheal syndromes of *B. cereus* infections, three protein enterotoxins are implicated, the hemolytic hemolysin B (Hbl), the non-hemolytic enterotoxin (Nhe) and the cytotoxin K (CytK) [13,14]. Even if associated symptoms in some cases can be mild and self-limited, like abdominal pain, watery diarrhea and nausea, lethal cases have also been reported [15].

#### 2.1.2. Structure of Hbl, Nhe and CytK

Similar structure and functionality characterize Hbl and Nhe enterotoxins as each of them is composed of three different protein compartments, resulting in a membrane attacking complex that leads to lysis of target cells. More specifically, Hbl consists of L2, L1 and B proteins produced at an equivalent ratio, while Nhe is composed of NheA, NheB and NheC proteins produced at a ratio close to 10:10:1 (NheA: B: C). Hbl L2 shows sequence homology to NheA, Hbl L1 to NheB and Hbl B to NheC. For exhibiting their maximal enterotoxic activity, the synergistic function of all three proteins is necessary. Concerning Hbl, X-ray crystallography results revealed a long α-helical bundle and a small α/β head domain to the toxin structure. Based on the similarity of the Hb1 structure with the structure of *E. coli* hemolysin E (HlyE, ClyA, SheA), pore formation was suggested as a mode of action for Hbl. Nhe also belongs to the pore formation toxins, however, its structure is distinct from the structure of Hbl, while it also lacks hemolytic activity [16]. CytK (34 kDa), or alternatively called hemolysin IV, is a single component, β-barrel channel-forming toxin [17]. Two different forms of CytK have been investigated, CytK 1 and CytK 2. CytK 1 was found to be five-fold more toxic to human intestinal Caco-2 and Vero cells than CytK 2 is [16].

#### 2.1.3. Pathogenicity of Hbl, Nhe and CytK

In their structure, all *B. cereus*-related enterotoxins contain secretory signal peptides, a fact that indicates their secretion by the general secretory (Sec) pathway [17]. To date, for Hbl and Nhe, no cell membrane receptors have been identi-fied so far [17]. However, some studies reported evidence of Nhe cellular activity which begins with the binding of NheC and NheB proteins. After binding, NheC and NheB oligomers get attached to the cell membranes to form prepores and finally NheA becomes associated with the NheB-C complex and penetrates the lipid bilayer leading to cell death [18]. Based on these studies, pore formation relies on a stepwise, sequential binding of NheC, NheB and NheA and, Hbl B, Hbl L1, and Hbl L2, respectively [19]. According to the significant amino acid sequence similarity between Nhe and Hbl, in addition to the fact that crystal structure of Hbl B resembles the well-known cytolysin A, it is likely that Hbl, like Nhe, also belongs to the α pore-forming toxins (PFT) family [18]. To achieve the optimal pore formation and the maximum cytotoxicity at the target cell surface, specific binding order as well as a specific concentration ration is needed from each enterotoxin separately.

CytK was recently described, therefore investigations about its pathogenic mechanism are still limited. However, in vitro studies have demonstrated its ability to form pores in the epithelial cells and therefore result in fluid release and necrosis due to epithelial cell destruction. Based on their ability to form pores in phospholipid membranes, it was believed that it is unlikely that receptors are absolutely necessary for binding and lysis by CytK [17]. To conclude, the pathogenicity and relation of all three enterotoxins with diarrheal effects is based on their ability to damage the integrity of the plasma membrane of the small intestine by forming pores that allow influx of Ca^2+^, Na^+^ and efflux of K^+^ and ATP and provoke cell death (Figure 1A).

### 2.2. C. perfringens and Related Enterotoxins

*C. perfringens*, a Gram-positive, spore-forming and anaerobic bacterium, which is one of the components of the normal gastrointestinal (GI) tract microbiota of both humans and animals. Except humans and animals, it can be also found in soil, food and sewage. The virulence of *C. perfringens* is attributed to the production of different toxins depending on the different strains. Its presence is associated with various histotoxic, enteric and enterotoxemic diseases due to this large production of enterotoxins, including traumatic gas gangrene, foodborne illnesses and enteritis necroticans [20].

*C. perfringens* strains are classified in five different types (A–E) according to the presence of encoding genes for alpha (α), beta (β), epsilon (ε) and iota (ι) toxins [20]. Recently, this classification system was revised and expanded with two more bacterial strains, type F strains, producing *C. perfringens* enterotoxin (CPE) and type G strains, producing necrotic enteritis B-like toxin (NetB) [21]. In this review, we focused on *C. perfringens* beta toxin (CPB) and CPE enterotoxins provoking human intestinal diseases.

#### 2.2.1. *C. perfringens* Beta Toxin (CPB)

##### Epidemiology of CPB

*C. perfringens* beta toxin (CPB) is the causative agent of foodborne necrotizing enterocolitis in humans, produced by type C strains. Additionally, termed as Pig-bel, necrotizing enterocolitis was historically more related to the Highland of Papua New Guinea and it was mainly caused by the ingestion of insufficiently cooked pork. Disease symptoms include serious bloody diarrhea, abdominal pain, distention and emesis [22].

##### Structure of CPB

CPB is expressed as a 336-amino-acid single polypeptide belonging to the β-PFT family. During CPB secretion, a 27-amino-acid signal sequence is removed, leading to the formation of the mature toxin (35 kDa). Purified CPB is highly sensitive to protease treatment and is thermolabile [23]. As a member of the β-PFT family, CPB shares sequence homology with other toxins, including *C. perfringens* delta toxin (43% identity) and several *S. aureus* toxins, including alpha toxin (28% identity) [24].

##### Mode of Action of CPB

CPB is expressed as a prototoxin that includes a signal sequence of 27 amino acids, which is removed upon toxin secretion, leading to a mature protein. CPB selectively binds to the ATP-gated P2X7 receptor on the plasma membrane. This binding is related to ATP release from the target cell through the ATP Pannexin 1 channel. ATP release facilitates CPB to get oligomerized and form functional pores. Through these pores, the efflux of intracytoplasmic K^+^ (iK^+^) and the entry of Ca^2+^, Na^+^ and Cl^−^ is allowed. On the other hand, influx of Ca^2+^ is associated with calpain activation and necroptosis (Figure 1B) [21].

#### 2.2.2. *C. perfringens* Enterotoxin (CPE)

##### Epidemiology of CPE

CPE represents a very important cause of different human illnesses, including foodborne and non-foodborne GI diseases [21]. CPE is mainly produced by type F CPE-positive strains, however, being produced by type C and D strains is also common [21]. Specifically, CPE-positive type F strains are the main etiological factor of *C. perfringens* type F food poisoning (previously referred as *C. perfringens* type A food poisoning), which is ranked as the second most frequent foodborne illness in most developed countries. In the USA, around one million food poisoning cases are estimated each year due to *C. perfringens* infection [25]. Symptoms like diarrhea and abdominal cramping caused by type F food poisoning have found to be highly associated with the presence of CPE and under predisposing conditions, CPE can also be fatal. In addition to food poising, CPE-positive type F strains cause 5 to 15% of all non-foodborne human GI diseases, such as, antibiotic-associated diarrhea and sporadic diarrhea. Additionally, CPE-positive type C strains are also responsible for some incidences of enteritis necroticans, a form of inflammatory gut disorder, even if the beta toxin constitutes the primary role in the pathogenesis of this disease [26].

##### Structure of CPE

CPE is a pore forming two-domain protein composed of a single chain polypeptide of 319 amino acids (35 kDa). The *C*-terminal domain (C-CPE, residues 184–319) is the receptor-binding domain that recognizes and binds to claudins as the receptors of the toxin. The *N*-terminal domain is involved in oligomerization and pore formation that disrupt plasma membrane, leading to cell death. A specific region of the *N*-terminal domain (residues 80–106) termed as TM1, consists of amino acids that resemble the β-hairpin loops, which are known to mediate membrane insertion and pore formation of several bacterial pore-forming toxins. This ability of the TM1 region largely corresponds to an α helix located in the *N*-terminal domain of CPE. Consequently, the α helix likely unfolds into a β-hairpin loop during pore formation and membrane penetration [27].

##### Mode of Action of CPE

Pathogenicity of type F by the *C. perfringens* strain begins with the consumption of food, contaminated with vegetative cells. In severe cases when the ingested food is highly infected, bacteria can survive upon the exposure of gastric acid and pass into the intestines. The *cpe* gene is located either in plasmids or in the chromosome and CPE expression may only happen during sporulation. After colonization of *C. perfringens* in the intestine, vegetative cells commit to sporulation and thus to the production of CPE. When sporulation is completed, the mother cells are lysed, and CPE is released into the intestinal lumen. Thereafter, CPE binds strongly to its cellular receptors that are several members of the claudin family. Claudins have been identified as the CPE-binding host proteins that play a critical role in maintaining the normal barrier and forming the backbone of tight junctions in the cell [28]. Especially, claudins 3, 4, 6, 8 and 14 are identified as CPE receptors among which claudins 3 and 4 exhibit the strongest binding. Additionally, claudins are composed of four transmembrane domains, a short C-terminal tail and two extracellular loops termed as ECL-1 and ECL-2. Both ECL-1 and ECL-2 regions were found to be necessary for CPE binding. Once CPE binds to claudin receptors, a “small complex” (90 kDa) is composed that is unable to trigger cytotoxicity. However, when several small CPE complexes are oligomerized, a prepore is formed on the plasma membrane surface that results in the construction of a “large complex” (450 kDa) also named as CH-1 [29]. At this CH-1 structure, β-hairpin loops assemble into a β-barrel structure that can quickly insert into membranes and thus form a cation-permeating pore [30]. Consequently, this CPE pore allows the influx of calcium that activates calpains, leading either to caspase-3 activation and apoptosis in low CPE doses or to necrosis at high CPE doses and excessive calcium influx (Figure 1C).

### 2.3. S. aureus and Staphylococcal Enterotoxins (SEs)

#### 2.3.1. Epidemiology of *S. aureus* and SEs in Foodborne Poisoning Associated Diarrhea

*S. aureus* is a Gram-positive, cocci-shaped bacterium that tends to form clusters usually described as “grape-like”. It is a facultative, coagulase positive microorganism that can grow between 18 and 40 °C and can ferment mannitol. In humans, it usually resides on the skin and on mucous membranes, especially in the anterior nares. It is estimated that *S. aureus-*related infections are one of the most common infections in humans, including gastroenteritis, skin and soft tissue infections, pulmonary infections, toxic shock syndrome, urinary tract infections, meningitis. According to the different strains of *S. aureus* and to the site of infection, *S. aureus* can cause intrusive infections and toxin-regulated diseases. The SEs are potent gastrointestinal toxins and constitute one of the most threatening virulence factors of *S. aureus* for human health [31].

SEs belong to a family of more than 20 different staphylococcal and streptococcal exotoxins with similar functionalities and sequence homology, mostly produced by *S. aureus*. SEs are characterized as pyrogenic enterotoxins and are related to severe human diseases, including food poisoning and toxic shock syndrome. Symptoms of infections related to SEs include diarrhea, vomiting, abdominal pain, cramps and nausea [32]. Centers for Disease Control estimated that SEs were associated with 80 million cases in USA, resulting in 325,000 hospitalizations and 5000 deaths [33]. Among foodborne diseases, staphylococcal-related ones are the second most prevalent. The main cause of this high incidence is attributed either to the insufficient pasteurization of the original product source or due to the contamination that may occur during preparation and handling by individual carriers of the pathogen [34].

#### 2.3.2. Structure of SEs

SEs are globular single-chain proteins broadly classified as SAgs. To date, at least 20 distinct staphylococcal SAgs have been identified, including SEs A through V and toxic shock syndrome toxin-1 (TSST-1) [32]. Regarding each enterotoxin subtype, the molecular weight (MW) of the toxins ranges from 20 to 30 kDa with an estimated mature protein length of 220–240 amino acids. It has been estimated that besides the variability of SAgs’ primary amino acid sequence, they share similar three-dimensional structures [32]. X-ray crystallography results revealed that their quaternary conformation is composed of an α-helical and β-strand structural complex, which creates an ellipsoid protein shape. A common characteristic of those enterotoxins is that they are made up of two unequal domains, stabilized by close packing, in the middle of which a cysteine loop structure is occasionally found, forming a disulfide bridge. When present, the disulfide bond is linked with the amino terminal part of the small protein domain forming an intervening variable loop. On the other hand, the large C-terminal domain is a β-grasp motif of a four- to five-strand β-sheet that packs against a conserved α-helix [35]. For antigen recognition, SAgs interact with major histocompatibility complex (MHC) class II molecules and with T-cell receptors (TCR). For the TCR-binding site, the N-terminal extension has a substantial role, while at least two distinct binding sites have been recognized for MHC II molecules. The first one is a common, overlapping, generic binding site involving the invariant α-chain of MHC II and the second is a high-affinity, zinc-dependent binding site on the polymorphic β-chain [36].

#### 2.3.3. Mechanism of Action of SEs in Gastro-Intestinal Inflammatory Injury

Food poisoning with SEs results in inflammatory changes in the gastrointestinal tract. In vitro and ex vivo studies proposed that in order to provoke GI inflammatory injury, SEs first cross the intestinal epithelial barrier and in their intact form, can bind to MHC class II molecules and subsequently attract CD4^+^ T cells. Thereafter, a strong production of pro-inflammatory cytokines and chemokines occurs, including IL-6, IL-8 and MCP-1. Secretion of MCP-1 may lead to increased chemotaxis of CD4^+^ T cells, macrophages and dendritic cells (DC) from gut-associated lymphoid tissue (GALT) to the inflammation site in GI mucosa. These interactions of SEs with MHC class II may result in hyper activation of professional (macrophages, DC) and non-professional (myofibroblasts) antigen -presenting cells and T cells. Thereafter, an excessive proliferation of CD4^+^ T cells is provoked, along with an outburst of pro-inflammatory cytokine and chemokine secretion leading to superantigens-mediated acute inflammation and shock (Figure 1D) [32].

### 2.4. C. difficile and Related Enterotoxins (TcdA, TcdB and CDT)

*C. difficile*, a Gram-positive bacillus, is an anaerobic, spore-forming intestinal colonizer. Studies have shown that *C. difficile* is the primary global cause of nosocomial antibiotic-associated diarrhea and pseudomembranous colitis [37]. From spore ingestion to symptoms’ occurrence, the time can vary, but typically is short (around 4 weeks). *C. difficile* is characterized by genetic diversity including both non-pathogenic and pathogenic strains. For the development of a clinical infection, successful germination of *C. difficile* spores resulting in toxin production is needed [38]. Symptoms of diseases related to *C. difficile* infections (CDI) may range from mild diarrhea to life-threatening complications such as intestinal perforation, toxic megacolon and pseudomembranous colitis [38]. To date, 7 to 10,000 hospitalized patients have been reported with CDI in Europe [39] while in USA, 14,000 deaths are estimated each year caused by hospital-associated CDI [40].

#### 2.4.1. *C. difficile* Toxin A (TcdA) and Toxin B (TcdB)

##### Epidemiology of TcdA and TcdB

*C. difficile* toxin A (TcdA) and *C. difficile* toxin B (TcdB), produced by the pathogenic strains of *C. difficile*, are both efficiently expressed during exponential and stationary phases of bacterial growth [41]. Acting as glucosyltransferases, TcdA and TcdB have been identified as the major virulence factors of C. difficile infections and are among the largest bacterial toxins reported since now.

##### Structure of TcdA and TcdB

TcdA and TcdB, two of the largest clostridium enterotoxins, share 48% of their amino acid sequence containing 2710 and 2366 amino acids, respectively. Structures of TcdA (308 kDa) and TcdB (270 kDa) are separated into four domains according to their functionalities based on an ABCD model. For both toxins, the biologically active A domain harbors the glucosyltransferase activity of the toxins and it is located at the N-terminal part, also called N-terminal glucosyltransferase domain (GTD). After the A domain, the C (cutting) domain follows that possesses the protease function. Finally, the D domain arranges the delivery of the toxin or part of it inside the cytosol of the host cell, while at the C-terminal part, the B domain is located and involved in receptor binding. Even if the full-length toxin has not been crystallized yet, single-particle analysis of domains A, C and D allow a nearly complete view of these toxins [42].

##### Mechanism of Action of TcdA and TcdB

As noted previously, TcdA and TcdB are structured following a multi-domain ABCD model with the goal to inactivate GTPases. Their binding domain is characterized by repetitive sequences, called combined repetitive peptides (CROP), and is located at their C-terminal end, being responsible for toxin attachment on the host cell membrane with different receptors. For TcdA, sucrose-isomaltase (SI) and glycoprotein 96, have been identified as plasma membrane receptors both expressed on human coloncytes [43,44]. Two other receptors, chondroitin sulfate proteoglycan 4 (CSPG4) and poliovirus receptor-like protein (PVRL3), have been identified as colonic epithelial receptors for TcdB [45,46]. Additionally, for both toxins glycan binding sites have been identified, indicating that more than one receptor is used for cell binding and entry [42]. After attachment to their receptors, TcdA and TcdB are internalized into host cells to disturb their function. Despite their homology, both toxins follow different entry mechanisms often directed by the cell surface receptors [37]. TcdA uses a clathrin-independent route for entry, controlled by PAC-SIN2 [47], while TcdB uses a clathrin-mediated endocytic pathway [48]. Once entering the cell, the toxins reach the acidified endosomes, where they undergo a pH-induced conformational change between the CROP and cysteine protease domain region. Structural alterations of toxins lead to pore formation that enables the N-terminal GTD domain to penetrate the endosomal membrane and be released into the host cell cytosol upon translocation [49]. An autocatalytic cleavage event was found to be induced by inositol hexakisphosphate (InsP_6_), a highly charged molecule abundantly found within mammalian cells [50]. At the end of their autocatalytic process, TcdA and TcdB glucosylate several members of the Rho subfamily (Figure 2A). The glucosylation of Rho proteins inhibits various cellular functions (e.g., epithelial barrier function, transcription) and signal pathways (e.g., cytokine production), processes crucial for the host [42].

#### 2.4.2. *C. difficile* Transferase (CDT)

##### Epidemiology of CDT

*C. difficile* transferase (CDT), the third *C. difficile* toxin, is secreted by a number of hypervirulent strains of *C. difficile* such as PCR-ribotypes 078 and 027 responsible for severe CDI outbreaks. Especially over the last decade, the frequency of patients infected by CDT-expressing strains has become increasingly prevalent in the human populations [51].

##### Structure of CDT

CDT belongs to the family of binary ADP-ribosylating toxins and is composed of two domains, the mature enzyme component CDTa (48 kDa) and the binding component CDTb (98.8 kDa). The N-terminal part of the CDTa contributes to the interaction with the CDTb component, while the C-terminal part is involved in the enzymatic activity. Concerning the CDTb binding compartment, is separated into four domains according to their functionalities. Domain I (residues 0–257) constitutes the activation domain, domain II (residues 258–480) is responsible for membrane insertion and pore formation, domain III (residues 481–591) participates in the oligomerization of the toxin and the C-terminal domain IV (residues 592–876) mediates the binding with the receptor CDTb is activated by proteolytic cleavage of the N-terminal of domain I that allows oligomerization and formation of heptamers [42].

##### Mechanism of Action of CDT

CDT follows a similar cytotoxicity pathway as the rest of the binary enterotoxins. The attachment of the CDTb domain to the lipolysis-stimulated lipoprotein receptor (LSR) is the first step of its journey inside the host cell [52]. LSR is highly expressed in multiple tissues, including colon and small intestine [53]. CDTb can bind to LSR in two distinct forms. Either as a monomer, which is proteolytically processed and oligomerized after LSR attachment, or as an heptamer whose precursor form was proteolytically activated and oligomerized before the attachment with LSR. Based on these findings, proteolytic activation of CDTb is not required for receptor binding, although it is essential for oligomerization and subsequent intoxication of host cells. Afterwards, the oligomer-receptor complex and more specifically the N-terminus of activated CDTb, acts as a docking platform for the enzymatic active CDTa component. Once bound to CDTb, the holotoxin is internalized and transported to acidified endosomes inside cells. The acidic environment inside endosomes triggers conformational changes in heptamers leading to a transmembrane pore formation through which CDTa can be translocated across the membrane, and finally reach the host cytosol [37]. Productive translocation of CDTa mostly relies on host proteins, including heat shock proteins 70 and 90 (Hsp70 and 90) and cyclophilin A (CypA) [54]. Inside the cytosol, CDTa, as an ADP-ribosyltransferase, catalyzes the transfer of ADP-ribose from nicotinamide adenine dinuclotide (NAD+) to Arg-177 of G-actin monomer inhibiting its polymerization. Intoxication of CDT may lead to loss of actin-based cytoskeleton and high toxin concentrations can even lead to cell death. At low concentrations, rearrangement of actin cytoskeleton triggers the formation of microtubule-based cell protrusions, which form a network on the surface of epithelial cells that enhances pathogen adherence and colonization [55].

### 2.5. V. cholerae and Cholera Toxin (CT)

#### 2.5.1. Epidemiology of *V. cholerae* and CT

*V. cholerae* is a Gram-negative bacterium that belongs to the Vibrionaceae family and more specifically to the gamma subdivision of the phylum Proteobacteria. It is a highly motile, curved oxidase-positive, facultative anaerobe that ferments glucose, sucrose and mannitol. *V. cholerae* is the causative agent of cholera but only the strains that produce cholera toxin are associated with the disease. Infection with V. cholerae is caused by the ingestion of contaminated food and water, especially in areas with poor sanitation and hygiene [56].

To date, seven cholera pandemics have been officially recognized [57]. After being digested and survived in the acidic gastric environment, *V. cholerae* colonizes in the small intestine epithelium producing CT. Secretion of CT leads to an extensive discharge of watery stool with direct consequences of electrolytes’ loss, rapid dehydration, fall of blood pressure, vomiting and cramps in legs and abdomen [58]. In severe cases, rapid loss of fluid leads to hypovolemic shock and metabolic acidosis [59]. World Health Organization estimates 1.4 to 4.4 million cholera cases with death rates between 21,000 and 143,000 per year [60].

#### 2.5.2. Structure of CT

CT belongs to the family of AB_5_ toxins and is an 84-kDa heterogeneous protein made up by two subunits. The first subunit is the CTA active domain (28 kDa) with a length of 240 amino acids, while the second one is the CTB homopentameric non-toxic B domain (56 kDa) of 103 amino acids in length per each monomer. Initially, CTA is synthetized as a single polypeptide chain, however after post-translational modification, CTA1 (residues 1–192; 23 kDa) and CTA2 (residues 193–240; 5 kDa) fragments are generated and remain linked together with a disulfide bond [61]. The toxic activity of CT resides in CTA1, whereas CTA2 mediates the insertion of CTA into the CTB pentamer. The circular B-subunit homopentamer is responsible for the binding of the toxin to cells and is held together by approximately 130 hydrogen bonds and 20 inter-subunit salt bridges, which provide an outstanding stability to the pentamer. The CTB pentamer and the CTA2 fragment are non-covalently linked except the sequence of the last four amino acids (lysine-aspartate-glutamate-leucine) at the carboxy-terminal of CTA2 that pro-trudes from the associated toxin and is not engaged in interactions with the pentamer [62].

#### 2.5.3. Trafficking and Mechanism of Action of CT

The pathogenic journey of CT inside epithelial cells, after *V. cholerae* colonization, begins with the attachment of the toxin to the GM1 ganglioside: β-D-Gal-(1→3)-β-D-GalNAc-(1→4)-[α-D-Neu5Ac-(2→3)]-β-D-Gal-(1→4)-β-D-Glc-ceramide. GM1 has been identified as the receptor of CT that facilitates its internalization inside the host cell membrane of epithelial cells [63]. For each B monomer there is one binding pocket for the GM1 receptor resulting in a total of five binding sites. The attachment of the toxin with the receptor mostly involves Gal and sialic acid, the two terminal sugars of GM1 [64]. After being attached, the GM1-toxin complex internalizes inside the cell from the plasma membrane to early endosomes following different entry routes [65]. Previous studies on distinct types of cells have reported various uptake mechanisms, such as, caveolae-dependent [66], clathrin-dependent [67], or non-caveolae/non clathrin-mediated pathways [68]. Regardless of the mode of entry, the toxin internalizes in early endosomes and travels to the transGolgi network following a retrograde transportation pathway [69]. The appearance of the last four amino acids residues (Lys-Asp-Glu-Leu) at the C-terminal of the A2 chain, termed as KDEL in the one-letter code, has been reported as an endoplasmic reticulum (ER) targeting motif, and its presence indicates the trafficking of the holotoxin to the ER [70]. By entering the lumen of ER as a holotoxin, the A subunit is proteolytically cleaved between residues Arg192 and Ser193 by a bacterial endoprotease to the wedge-shaped A1 subunit and to the elongated A2 subunit. The two subunits remain linked with an intrachain disulfide bond (Cys187 = Cys199) until the trafficking of the A1 subunit inside the cytosol, where the disulfide bond is reduced by the purified protein disulfide isomerase [69]. Through SEC61 channel, the A1 chain reaches the cytosol where it catalyzes ADP-ribosylation of the a-subunit of the stimulatory G protein (Gs). This post translational modification occurs by transferring an ADP-ribose from nicotinamide adenine dinuclotide (NAD^+^) to an arginine residue of the protein. Being ribosylated, the stimulatory G protein in its guanosine triphosphate (GTP)-bound form, activates adenylyl cyclase (AC), which subsequently induces the conversion of ATP into cyclic AMP (cAMP). Elevated levels of intracellular cAMP in crypt cells activate protein kinase A (PKA) that phosphorylates the cAMP-dependent chloride channel (CFTR). Through cystic fibrosis transmembrane conductance regulator (CFTR), an enhanced efflux of electrolytes and water occurs in the intestinal lumen with a direct result of the production of a large volume of watery diarrhea (Figure 2B) [61].

### 2.6. E. coli and Related Enterotoxins

*E. coli*, a Gram-negative bacterium, is an aerobic, non-spore-forming bacillus that belongs to the family of Enterobacteriaceae. *E. coli* strains are classified into different categories according to the unique interactions with eukaryotic cells. Some of the most known *E. coli*-related enterotoxins are the enterotoxigenic *E. coli* (ETEC), the enteropathogenic *E. coli* (EPEC), the enterohemorrhagic *E. coli* (EHEC), the Shiga toxin-producing *E. coli* (STEC) and the uropathogenic *E. coli* (UPEC) [56]. Based on its pathogenic potential, *E. coli* is divided into non-pathogenic groups in which it acts as a commensal member of the gut microbiota and to pathogenic groups. *E. coli* is associated with a wide range of infections including intestinal pathologies (e.g., gastroenteritis) and extra-intestinal pathologies (e.g., urinary tract infections, neonatal meningitis and bacteremia). Even if some of those infections can be very common like urinary tract infections, some others like hemolytic uremic syndrome (HUS) can be associated with high morbidity and mortality [71].

#### 2.6.1. *E. coli*, LTs and STs

##### Epidemiology LTs and STs

Enterotoxigenic *Escherichia coli* (ETEC) is the major causative agent of induced diarrhea mainly associated with the production of several enterotoxins including LTs and STs. Approximately 840 million infections of ETEC-induced diarrhea are estimated each year as well as around 3,800,000 deaths worldwide [72].

LTs produced by *V. cholerae* and *E. coli* are members of a family of structurally identical proteins that provoke diarrheal symptoms in humans. The family is composed of two major subcategories according to their biochemical, genetic and immunological characteristics. The Type I subcategory is comprised of CT from *V. cholerae* and LT-I enterotoxin (or LT) from *E. coli*. Type II consists of LT-IIa enterotoxin and its partially cross-reacting antigenic variant, LT-IIb, both produced from *E. coli*. The A and B polypeptides of both CT and all LTs are synthesized inside the cytoplasm as precursors, including an N-terminal signal sequence. From the cytoplasm, they get secreted into the periplasmic space where they assemble spontaneously to the holotoxin molecules by removing the signal sequence from their structure [73]. In addition to *V. cholerae* and *E. coli*, other enteric pathogens like *Aeromonas hydrophilia* and *Plesiomonas shigelloides* have also been found to produce cholera toxin-like LTs. Along with structural similarities of LT-I and LT-II with CT, LTs also follow identical intoxication pathways inside host cells. However, minor differences in terms of receptor type, C-terminal motifs and ADP-ribose acceptors, distinguish toxins’ pathogenicity routes [73].

Strains of ETEC-producing STs are ranked among the top eight enteropathogens that cause diarrhea, accounting for a mortality rate of 3.2% in 2016. Especially in children, ETEC infection may lead to long term consequences including stunted growth, chronic gut inflammation and impaired cognate development [74]. Except for their distinct role in diseases, heat-stable enterotoxin type a (STa) is characterized as methanol soluble and protease resistant, while heat-stable enterotoxin type b (STb) is exactly the opposite. This review focuses mostly on STa as our review concerns human-related diseases.

##### Structure of LTs and STs

LTs belong to the family of AB_5_ toxins and are mostly related to CT both in terms of structure and of function. Therefore, LTs are oligomeric toxins (86 kDa) composed of two main subunits, the A subunit (28 kDa) and the pentameric B subunit (58 kDa). Enzymatic catalytic activity of LTs is attributed to free LTA, while LTB pentamer is responsible for the entry of the toxin into host cells [75]. LTs, like CT, bind to the GM1 receptor with high affinity, with five binding sites per each B pentamer. Except GM1, LTs can alternatively bind to blood group A and B determinants through a novel binding site which is located in a distinct position from the GM1 binding site [72].

STs constitute a family of low-molecular-mass peptide toxins, divided into two categories, Sta and STb. Sta is more associated with diarrhea induced in humans and is subcategorized into two variants, STh, a 19 amino-acid protein (2 kDa), found in human ETEC strains and STp, an 18 amino-acid protein (2 kDa), isolated from human and porcine strains [76]. STh and STp are almost identical since they include 14 similar amino acid residues. In addition, all of them contain a cysteine-rich core that is crucial for the expression of their biologic activity. Inside the core, cysteine residues are all arranged in disulfide bridges whose integrity is an essential requirement for the maintenance of toxin conformation [77]. On the other hand, STb is a 48 amino-acid protein (5.1 kDa) which is virulent only against animals [76].

##### Trafficking and Mechanism of Action LTs

To begin cell intoxication, the binding domain of LTs binds to the glycolipid receptors in plasma membranes and the toxin-receptor complex enters inside the cell through endocytosis. Following a retrograde transportation pathway, the holotoxin travels to the Golgi apparatus and subsequently to the ER. Different C-terminals motifs of the A2 tail are recognized by a luminal ER membrane protein (ERD2) that can retrieve toxins from the Golgi apparatus. The four last amino acid residues (Lys-Asp-Glu-Leu) of CT and LT-IIb are termed KDEL in the one-letter code while for LT-I and LT-IIa are termed RDEL (Arg-Asp-Glu-Leu). After proteolytic cleavage of A subunit, reduced A1 crosses the ER membrane lipid bilayers and gains access inside the cytosol by exploiting the Sec61 machinery. Despite the fact that normally proteins that are exported from the ER by similar processes are targeted for degradation by the proteasome, the scarcity of lysines in the active part of toxin, prevents toxins from ubiquitination and degradation. Inside the cytosol, the expression of A1 toxic effects begins with the ADP-ribosylation of an arginine residue in the stimulatory alpha (Gsα) subunit of the heterotrimeric regulatory G protein complex. A1 subunits may use different substrates as ADP-ribose acceptors including arginine and other guanidine compounds, such as, agmatine. The most substantial difference among LTs members is the much lower activity of the LT-II toxins that use agmatine as acceptor. Inactivation of intrinsic GTPase activity of Gsα stimulates AC that subsequently leads to elevated levels of cAMP. As a consequence, accumulation of cAMP, activates PKA-dependent pathways that stimulates Cl^−^ secretion by phosphorylation of the CFTR channel resulting in the loss of fluid and electrolytes inside the intestinal lumen (Figure 2B). Disturbance of electrolyte and water balance inside the cell provokes severe volumes of watery diarrhea [73].

##### Trafficking and Mechanism of Action of STs

Intoxication pathway of STa begins with toxin binding to the guanylate cyclase C (GC-C) receptor in the membrane of the small intestinal epithelial cells (IECs). Through the activation of the intracellular catalytic domain, Sta causes the hydrolysis of GTP that induces the accumulation of intracellular cyclic GMP (cGMP) levels. Increased levels of cGMP activate cGMP-dependent protein kinase II (PKGII) and inhibit phosphodiesterase 3 (PDE3) which in turn activates cAMP-dependent PKA. In their activated forms, PKGII and PKA phosphorylate and open the CFTR Cl^−^ channel through which, Cl^−^ and HCO_3_^−^ are released into the intestinal lumen. Finally, PKA can inhibit Na^+^ reabsorption by phosphorylating Na^+^/H^+^ exchanger 3 (NHE3) [74]. The inhibitory and activation effects of STa on the ionic channels finally leads to electrolyte imbalance and cell apoptosis (Figure 2C).

#### 2.6.2. *E. coli* and Stxs

##### Epidemiology of Stxs

The prototypical member of Stx family was first released from *Shigella dysenteriae* serotype1 followed by a Shiga-like toxin type, almost identical to the original one, produced by Stx-producing *E. coli* (STEC) [78]. Stx-producing bacteria enter the intestine through ingestion of contaminated food and water, which may lead to various diseases ranging from asymptomatic carriage to hemorrhagic colitis and HUS [79]. Annually, STEC causes 2,801,000 acute illnesses worldwide that lead to 3890 cases of HUS [80]. Even if the majority of patients can recuperate from uncomplicated diarrheal disease within a week, 6 to 9% of the patients develop HUS with devastating consequences, such as, thrombocytopenia, microangiopathic hemolytic anemia, and acute renal injury. Especially in children, STEC infection is the primary cause of acute kidney failure along with the high incidence of such infections in the elderly population [79].

Stx1 and Stx2 are the two main types of Stx family released from Stx-producing bacteria. Each type of toxin is separated to several subtypes differentiated according to their amino acid diversity. Stx1a, Stx1c and Stx1d are attributed to Stx1 whereas Stx2a-Stx2g to Stx2 [81]. Genes coding for Stx (*stx* gene) are embedded within the genomes of lambdoid bacteriophages instead of the actual bacterial genome, and remain silent during the lysogenic cycle but Stx production is initiated after lytic cycle induction [82].

##### Structure of Stxs

The structure of Stxs corresponds to an AB_5_ protein model. The A subunit (~32 kDa), non-covalently linked to the homopentameric B subunit (~7.7 kDa), is responsible for the enzymatic activity of the toxin. In order to be enzymatically active, the A subunit is composed of two separate fragments, the A1 (27.5 kDa) and the A2 (4.5 kDa) fragments that remain linked with a disulfide bond (Cys242 = Cys261). The A1 fragment possesses the *N*-glycosidase activity of Stxs, while the C-terminus of the A2 fragment mediates the binding with the B compartment. On the other hand, the B subunit binds to the glycosphingolipid Gb3 (α-D-Gal-(1→4)-β-D-Gal-(1→4)-β-D-Glc-ceramide) receptor that is identified as the native receptor of Stxs that facilitates toxin host internalization. For each B monomer, three binding sites have been identified resulting in 15 total binding sites for every toxin unity [83].

##### Trafficking and Mechanism of Action of Stxs

Stxs are released into the intestinal lumen following different endocytic pathways according to the presence or absence of glycosphingolipid Gb3. Gb3 acting as Stx receptor is located on the host cell membranes. In tumorigenic colon cancer cells where Gb3 is expressed, Stx exploits the binding with the Gb3 receptor to internalize inside host cells. At human IECs where Gb3 is not expressed, Stx follows Gb3-independent pathways, such as, macropinocytosis. Generally, IECs utilize macropinocytosis, a clathrin-independent mechanism, for the internalization of large molecules [84]. Previous studies have proven the association of Stxs with actin-coated macropinosomes in the intestinal epithelium and their subsequent transportation from apical to basolateral surfaces [85]. Regardless of the endocytic pathway, the toxin is transferred to the *trans*-Golgi network followed by the ER lumen retrograde intracellular transportation [86]. During trafficking to the ER, furin cleaves the A subunit into A1 and A2 fragments which remain linked via a disulfide bond that is reduced once the toxin enters the ER [79]. Through the ER-associated protein degradation pathway, unfolded A1 toxin exits the ER and enters the cytoplasm where it cleaves an adenine base of 28S ribosomal RNA of eukaryotic ribosomes. As a result, protein chain synthesis in the cell is inhibited, since the injured ribosome is no longer associated with the elongation factor-dependent amino-acyl tRNA [81]. The presence of the A1 polypeptide inside the cytoplasm not only inhibits the protein synthesis, but also activates several stress response pathways, such as ribotoxic stress and the ER stress response [87] as depicted in Figure 2D. After entering the bloodstream, toxins aim to reach their targets, including the kidneys and the brain, leading to systemic complications and in severe cases to death.

## 3. Non-Digestible Oligosaccharides

NDOs are carbohydrate moieties composed of less than 20 monosaccharide building blocks linked via glycosidic bonds [88]. The number of monomeric sugars of every NDO structure determines the DP of each moiety, which subsequently may influence their anti-virulent behavior [89]. An overview of the basic structures of the NDOs that demonstrated a certain role in fighting against EPB and associated enterotoxins is depicted in Table 1. NDOs that exhibit anti-pathogenic/anti-virulence effects are alginate-oligosaccharides (AOS), chito-oligosaccharides (COS), fructo-oligosaccharides (FOS), galacto-oligosaccharides (GOS), human milk oligosaccharides (HMOs), isomalto-oligosaccharides (IMOS), mannan-oligosaccharides (MOS), pectic-oligosaccharides (POS) and xylo-oligosaccharides (XOS).

AOS are derived from the enzymatic depolymerization or the acid hydrolysis of alginate, a biopolymer present in the cell walls of brown algae [90]. Alginate contains two monosaccharide building blocks, (1→4)–linked β-D-Mannuronic acid (ManA/M) and (1→4) α-L-Guluronic acid (GulA/G). These can be either homogenously or heterogeneously linked forming homodimers (GG/MM) or heterodimers (MG/GM) [90].

COS are degraded products of chitin and of chitosan, produced after enzymatic and chemical hydrolysis [91]. Chitin is present on crustacean or arthropodic shells and contains a high proportion of β-(1→4)-linked N-acetylglucosamine (GlcNAc), while chitosan is present in cell walls of specific fungi and is mainly composed of the β-(1→4)–linked D-glucosamine (GlcN) [92]. The average molecular weight (MW) of COS is less than 3900 Da, while the DP is less than 20 [91].

FOS, also known as oligofructose or oligo fructan, are naturally found in higher plants, like fruits and vegetables and can be either obtained by plant extraction or by enzymatic manu-facture [93]. The chemical structure of FOS is composed of a linear chain of fructose units linked by β-(2→1) glycosidic bonds, terminated by a glucose (Glc) unit linked with an α-(1→2) glycosidic bond. Structures of FOS with a DP of more than 10 are termed as inulin [94].

GOS can be synthesized from lactose by a β-galactosidase enzyme in a reaction known as transgalactosylation [95]. Monosaccharide building blocks included in the structure of GOS are galactose (Gal) units (DP = 2–6), linked by different bonds such as β-(1→3), β-(1→4) and β-(1→6) glycosidic linkages terminate in a β-(1→4) linked glucose unit [96].

Since FOS and GOS resemble the oligosaccharides that are naturally present in human breast milk, several types of infant formula are supplemented with those two oligosaccharides to obtain the advantages of a breast-fed microflora [97].

HMOs constitute a key component of human milk and represent a group of structural and biological diverse and complex indigestible sugars. According to the different monosaccharide moieties of their structure, HMOs can generally be divided into neutral oligosaccharides, containing occasionally fucose (Fuc) units (fucosylated HMOs) and acidic oligosaccharides (sialylated HMOs), containing sialic acid units. The basic monosaccharide components of HMOs are Gal, Glc, Fuc and GlcNAc, or sialic acid [98].

IMOS are naturally present at low concentrations in honey and in fermented foods like in soy sauce. Alternatively, IMOS can be manufactured by an enzymatic process utilizing starch as the substrate. The main monosaccharides components of IMOS are Glc units linked with α-D-(1→6) glycosidic bonds, with a DP range from 3 to 6 [93]. MOS can be chemically synthesized or obtained from the outer cell-wall membrane of bacteria, plants or yeast [99]. Isolated structures of MOS are mostly composed of (1→2), (1→4) and (1→6) D-mannose linkages [100].

POS are obtained by the depolymerization of pectin, a plant complex macromolecule made up of several monosaccharides [101]. Pectins are most importantly composed of a linear backbone of α-(1→4)-linked D-Galacturonic acid (GalA) units that can be partially acetylated and/or methylated [102]. The linear structures of pectins, termed as “smooth” homogalacturonic regions are made up by GalA and are occasionally interrupted by rhamnose residues called “hairy” rhamnogalacturonic regions [102].

XOS are naturally found in bamboo shoots, fruits, vegetables, milk and honey and are formed by 2 to 10 xylose molecules, linked by β-(1→4) glyocosidic bonds [103]. Additionally, XOS can be produced chemically by direct enzymatic hydrolysis of xylan [104].

Structural characteristics of the mentioned NDOs play a crucial role in their mechanism of action against EPB and enterotoxins, as will be discussed in the next paragraph.

## 4. Different Effects of NDOs and SCFA against Enterotoxins and Enterotoxin-Producing Bacteria

### 4.1. The Effects of NDOs and SCFA against Enterotoxin-Producing Bacteria

NDOs are proven to have crucial roles in protecting the body from pathogenic bacteria. They play this role via at least two different pathways inducing both direct and indirect defenses against these pathogens. First, NDOs can also encounter toxigenic attacks through their direct interaction with EPB inducing anti-adhesive, anti-biofilm and anti-growth effects. Their anti-adhesive capability is based on the similarity of their carbohydrate backbone with the structure of EPB receptors on host cells. Furthermore, NDOs can disrupt or inhibit the formation of biofilms, an extracellular polymeric sub-stance (EPS) matrix that pathogenic bacteria develop as a protective mechanism [105]. Interestingly, NDOs can also exert antibacterial effects that directly result in growth inhibition of pathogenic bacteria, including enterotoxin-producing microorganisms [106]. Second, NDOs can modulate the microbiota balance by encouraging the growth of beneficial bacteria in the gut lumen. Several studies proved that this modulation will not only result in an increase in SCFA production, but also in a decrease in the essential sources and space that pathogenic bacteria use to proliferate inside the intestinal lumen. This indirect effect of NDOs on the proliferation of EPB was investigated in several studies, which are depicted in Table 2. Concerning the indirect mode of action, oligosaccharide substrates get fermented by the potential beneficial bacteria species, resulting in an increased production of SCFA [2]. SCFA are volatile fatty acids produced in the large bowel, structurally characterized by fewer than six carbons, existing both in straight and branched-chain versions. After being produced, SCFA are absorbed and used in various biosynthetic pathways by the host, constituting an energy source [107]. Oligosaccharide-induced microbiome composition can benefit host health and protect against EPB by microbiota-dependent mechanisms.

#### 4.1.1. Direct Mechanisms of Action of NDOs against EPB

Various bacteria found in the intestinal tract are able to interact with the digestive mucosa and produce virulence factors responsible for gastrointestinal or foodborne diseases. Among the diverse virulence factors, enterotoxins represent the most invasive way to affect target cells. One category of toxigenic bacteria that produce enterotoxins are colonizing bacteria (e.g., *E. coli*, *V. cholerae)* whose adherent factors (e.g., pili, fimbriae) permit the evasion of inhibitory microflora. The second category are bacteria (e.g., *B. cereus*, *S. aureus)* that can grow and secrete their toxins in different environments, such as, food, leading to the digestion of pre-formed toxins that cause food intoxication. Finally, a third class of toxigenic bacteria (e.g., *C. difficile*, *C. perfringens)* can enter the digestive tract and grow in the intestinal lumen under certain conditions such as antibiotic treatment, to overcome the inhibitory effects of resident microflora [6]. Even if enterotoxins constitute one of the major therapeutic targets against EPB, several additional targets (e.g., adherence, biofilm formation) have also been investigated. NDOs have shown promising therapeutic potentials through their interaction with EPB. Direct anti-pathogenic capabilities of NDOs against EPB target both bacterial and enterotoxin toxicity pathways and mostly rely on the structural similarity of free NDOs with the carbohydrate patterns presented on the host cell surface [106]. A concise overview of the direct effects of NDOs against EPB is depicted in Table 3.

##### *E. coli* 

A great number of NDOs can exhibit antibacterial effects against different pathogenic strains of *E. coli*.

The anti-adhesive capabilities of NDOs, such as GOS are mostly related to molecular mimicry mechanisms in which free oligosaccharides resemble the structure of host cells carbohydrate based receptors [127]. The way that *E. coli* binds to epithelial cells mostly relies on the adhesins found on the fimbriae appendage of its structure. Different fractions of HMOs, including neutral, fucosylated and acidic fractions are known to inhibit adhesion of *E. coli* to epithelial cells. Fractions of neutral HMOs inhibit the adherence of an *E. coli* strain which is P-fimbriated and specifically recognizes galabiose or galactose structures on host cell surface [114]. Fucosylated HMOs, a subcategory of neutral HMOs decorated by fucosyl residues, inhibit Enteropathogenic *E. coli* (EPEC) adhesion on HEP-2 monolayers [115]. However, the great abundance of different molecular structures incorporated in HMOs mixtures makes it difficult to specify the effectiveness of a specific saccharide. Negatively charged sialylated HMOs can interact with oppositely charged elements on the epithelial cell exterior, showing inhibitory potential towards pathogenic species [116]. However, acidic HMOs can partially inhibit P and CFA fimbriae-expressing *E. coli* as the P-fimbrial lectin lacks affinity for sialylated oligosaccharides [114].

Anti-adhesive effects against *E. coli* are also exhibited by POS following a P-fimbriae-mediated inhibition mechanism [131]. However, although the POS structure lacks the exact α-Gal-(1-4)-β-Gal termini that P-fimbriated *E. coli* utilizes to adhere to epithelial cells, receptor mimicry is likely involved, but additional mechanisms remain unknown [140]. Furthermore, MOS can inhibit *E. coli* adherence to epithelial cells as the FimH domain of type I fimbriae, commonly found in *E. coli*, recognizes mannose patterns on host cells in order to be adhered. MOS can bind to the FimH domain and compete with the mannose patterns, inhibiting pathogenic adhesion through a receptor-mimicking function [129]. Finally, COS was also found to inhibit EPEC adhesion, while it did not exert the same capability against Verocytotoxin-producing *E. coli* (VTEC), showing target specificity [122]. Even if the reason for strain selectivity is currently undetermined, the anti-adhesive effect is presumably again a result of molecular mimicry, while the GlcNAc moiety on the cell surface receptor is recognized by the GafD adhesin of the G fimbriae containing *E. coli* strain and simultaneously constitutes one of the basic compartments of COS structure [122].

In addition to anti-adhesive capabilities of NDOs, other anti-pathogenic effects against *E. coli* have also been identified including bacterial cell membrane disruption, biofilm inhibition and radical scavenging. A great number of studies have shown that COS act as antibacterial agents via the inhibition of *E. coli* growth [123,124] The positively charged amino groups of chito-oligomers can bind to the negatively charged O-specific antigenic units of the *E. coli*, thereby blocking the nutrient flow leading to bacterial death due to nutrient depletion [126]. Additionally, AOS block pathogenic swarming and motility in *E. coli*, two substantial mediators for biofilm formulation [120]. POS can inhibit *E. coli* growth by scavenging free radicals like HO•, reacting with them and producing carbon dioxide radical anion (CO_2_•^−^). However*, E. coli* is inhibited less significantly by POS in comparison with *S. aureus* [130].

##### *V. cholerea* 

NDOs exhibit anti-adhesive and antibacterial effects against pathogenic *V. cholerea*. Among several NDOs, neutral high-molecular weight (HMW) HMOs and COS have shown such anti-pathogenic potential against *V. cholerea*, following receptor mimicking and membrane disruption mechanisms [116,134]. In order to achieve colonization in the small intestine, *V. cholerea* expresses the N-acetylglucosamine-binding protein A (GbpA), a nonspecific adhesin that facilitates attachment to the intestinal epithelium by specific binding to GlcNAc oligosaccharides [141]. GlcNac consists of a structural component of glycoproteins and glycolipids that are located on the IECs and on mucus. Except the abundance of GlcNaC on cellular moieties, it also constitutes a basic component of COS and HMOs [56]. Therefore, NDOs that include GlcNAc in their structure can mimic host cell receptors and compete for their binding with GbpA. Through a receptor-mimicry mechanism, the HMW fraction of HMOs was shown to inhibit the adhesion of *V. cholerae* to Caco-2 cells, thereby reducing its pathogenicity [116].

Furthermore, the antibacterial potentials of COS against pathogenic bacteria were related to the increased solubility and their cationic nature. The polycationic nature of COS enables them to adhere to Gram-negative bacteria, such as *V. cholerae*, creating a cationic oligosaccharide layer around them [142]. As it is adhered to the bacterial cell surface after prolonged exposure, COS can promote the leakage of proteinaceous and other intracellular constituents, resulting in cell swelling, cell lysis and inhibition of bacterial growth [143]. Consequently, this mechanism is an estimated explanation for the bactericidal activity of COS and NAc-COS against *V. cholerae,* possibly due to the lower MW and the higher solubility of chito-oligomers in comparison with the polymeric chitosan moieties [134].

##### *B. cereus* 

Among different antibacterial properties of NDOs, the anti-pathogenic effects of COS prevail against *B. cereus*. COS have shown to inhibit microbial cells either by interfering with the cell surface of bacteria or by blocking the transcription of RNA from DNA [134]. The cationic nature of chito-oligomers due to the positively charged NH_3_^+^ groups, enable them to bind with the peptidoglycan layers of Gram-positive bacteria, such as *B. cereus,* a Gram-positive bacterium. The cell wall of *B. cereus* is composed of peptidoglycan layers to which polycationic moieties (positively charged NH_3_^+^ groups) of chito-oligomers can bind. The binding that occurs leads to cell wall disruption, exposure of cell membrane to osmotic shock and secretion of intracellular substances that ultimately result in growth inhibition of Gram-positive bacteria, such as *B. cereus* [126].

##### *S. aureus* 

Multiple NDOs have shown antibacterial activities against *S. aureus*, a Gram-positive bacterium that is characterized by its biofilm formation and multi-drug resistance [144]. In order to encounter *S. aureus* escape mechanisms, the ability of COS to potentiate conventional antibiotics was investigated. Indeed, results have shown that COS enhanced the activity of several antibiotics possibly through the lysis of the bacterial cell wall [145]. Generally, anti-pathogenic features of COS are associated with a number of factors, including MW and degree of deacetylation (DD) [146]. COS with a MW less than 5000 can penetrate through the bacterial membrane in order to bind to bacterial DNA and inhibit RNA synthesis [147]. Bearing a positive charge, COS are also able to create an impermeable cationic oligosaccharide layer around the surface of *S. aureus* bacteria, thereby preventing the diffusion of metal ions and other nutrients, elements that are essential for bacterial proliferation, across the bacterial membrane [136].

Plant-based oligosaccharides, like POS, can also inhibit the growth and the adhesion of *S. aureus*. Characterized by redox activity, POS act as antioxidants, since they can efficiently scavenge free radicals. To eliminate pathological effects induced by free radicals such as HO•, POS react with HO• and produce the carbon dioxide radical anion (CO_2_•^−^), which is hypothesized to inhibit *S. aureus* [130]. However, since POS are characterized by great versatility, the antibacterial potential of CO_2_•^−^ needs to be further elucidated. Moreover, anti-adhesive effects of POS are mostly attributed to the high uronic acid content of their structure that results in higher ionic interactions among POS and pathogenic bacteria. The anti-adhesive mechanism of POS, derived from panax ginseng, was proposed against pathogenic Gram-positive bacteria, including *S. aureus* [137]. However, similar anti-adhesive effects were not exerted against beneficial and commensal bacteria, indicating that POS act in a strain-dependent manner.

Finally, HMOs and XOS are able to inhibit the growth of *S. aureus* by exhibiting anti-biofilm activity. Indeed, HMOs isolates from several donors significantly reduced biofilm production of MRSA, while the reductions ranged from 30 to 60% in comparison to the control [105]. Moreover, XOS demonstrated antibacterial activity against *S. aureus,* not only by inhibiting biofilm formation, but also by affecting cell membrane permeability and obstructing Ca^2+^-Mg^2+^- ATPase activity on the cytomembrane of *S. aureus* [138].

##### *Clostridium* spp.

Studies concerning the direct anti-pathogenic functionalities of NDOs against *Clostridium* spp. are very limited. To date, only FOS have been found to exert direct antimicrobial effects against *C. difficile*. More specifically, through an in vitro study FOS exhibited anti-adhesion potential towards several *C. difficile* strains on human epithelial cells [139]. Although the underlying mechanism of these anti-adhesive properties was not investigated, it was speculated that FOS possibly affect the surface proteins and adhesins of the bacteria. Additionally, 8% FOS was found to significantly reduce *C. difficile* biofilm formation [139]. The anti-biofilm effect of FOS may be correlated with its anti-adhesive effect since adhesion constitutes the primary step of colonization and biofilm formation.

NDOs can also fight indirectly against EPB, however since this is not the focus of the review, the indirect mechanisms will be shortly discussed in the next paragraph.

#### 4.1.2. Indirect Mechanisms of Action of NDOs and SCFA against EPB

NDOs and SCFA can maintain gut homeostasis through indirect mechanisms based on microbiota-dependent effects, such as antimicrobial activity of beneficial bacteria, as well as microbiota-independent effects related to barrier-protecting and immune-related properties. Initially, NDOs and SCFA, as their metabolites, can stimulate the growth of beneficial bacteria, which subsequently interfere with the maintenance of gut homeostasis and finally decrease the pathogenic effect of EBP. The most abundant beneficial bacteria are *Bifidobacterium* and *Lactobacillus* species, while the major byproducts of oligosaccharide fermentation incorporated in the family of SCFA, are acetate (mainly produced by *bifidobacteria*), propionate (produced by *propionibacteria* and *Bacteroidetes*) and butyrate (mainly produced by *Lachnospiraceae* and *Ruminococcaceae*) [148]. The defense role of the gut microbiota against EPB is based on several mechanisms, such as, antimicrobial activity and host immunity regulation induced by both beneficial bacteria and SCFA. Furthermore, gut microbiota acts against EPB by improving the intestinal barrier function of host cells and by reducing the luminal colonic pH due to the production of SCFA. Beyond the regulation of intestinal immunity and barrier function through microbiota-dependent effects, NDOs can promote gut immunity by their direct effects on specific immune and intestinal epithelial cells and also improve the intestinal barrier function by affecting epithelial tight junction proteins and goblet cell function. Since the main focus of this review is based on the direct mechanisms of action of NDOs and SCFA against enterotoxins and EPB, the indirect mechanisms of action of NDOs and SCFA against EPB will be shortly discussed in this review.

##### Antimicrobial Activity of Beneficial Bacteria

NDOs promote the growth of beneficial bacteria, such as, bifidobacteria or lactobacilli, which in turn exert anti-pathogenic capabilities. Beneficial bacteria can also interfere with the adhesion of bacterial pathogens and exert antimicrobial activity by inhibiting their growth [149]. FOS and inulin enhanced the antimicrobial activities of *Lactobacillus* spp. against pathogenic *S. aureus* and *E. coli*, an effect mostly related to the function of SCFA (acetate, propionate, isobutyric acid and butyrate) [110]. An increase in the growth of *Lactobacillus* and *Bifidobacterium* spp. was also induced by XOS that additionally suppressed the growth of *C. perfringens*. Moreover, FOS supplementation in combination with five different probiotics provoked growth inhibition of *E. coli* and *C. difficile*, although no mechanism was determined [108]. Finally, stimulation of different *Bifidobacterium* and *Lactobacillus* spp. growth by GOS and IMOS conferred protection against *C. difficile* infected mice [113].

##### Immunomodulation Activity

NDOs can exhibit anti-inflammatory effects that are likely to be driven by beneficial gut bacteria and their metabolites. Indeed, stimulation of *Bifidobacterium* growth by GOS, diminished the incidence of colitis leading to enhanced NK cell function and IL-15 production [150]. Several beneficial bacteria such as different *Bifidobacterium* spp. were also found to increase the levels of IgA-producing cells in the lamina propria, therefore stimulating the secretion of sIgA into the luminal mucus layers and preventing the colonization of bacteria in the epithelium [151]. Furthermore, via a cascade of signaling events, beneficial bacteria can promote the secretion and the production of anti-inflammatory cytokines, such as, IL-10 and TGFβ by T-regulatory cells [152]. Additionally, FOS can reduce intestinal inflammation and colitis incidences, mediated by the induced growth of intestinal lactic acid bacteria in the colon [153].

In addition to the regulation of gut immunity through microbiota-dependent effects, NDOs can promote intestinal immunity by their direct effects on specific immune cells and intestinal epithelial cells. Indeed, several in vitro studies demonstrated the effects of NDOs on cytokine and chemokine production and release by different intestinal epithelial cell lines exposed to inflammatory triggers. For example, 2′-fucosyllactose inhibited the induction of IL-8 caused by different strains of *E. coli* in T84 cells [154], while GOS prevented the secretion of IL-8 in Caco-2 cells [155]. Direct effects of NDOs on immune and epithelial cells are extensively reviewed by Yang et al. and Jeurink et al. [156,157].

##### Improvement of Intestinal Barrier Function

The disruption of epithelial barrier integrity constitutes one of the major pathological effects of EPB, and different studies describe that NDOs may substantially contribute to the protection of the epithelial barrier. Several NDOs improve intestinal epithelial barrier integrity by stimulating the growth of beneficial bacteria [158]. Multiprotein complexes, termed as tight junctions (TJ), tightly connect epithelial cells to their neighbors in order to control paracellular permeability and transepithelial transport [159]. FOS supplementation in mice decreased intestinal permeability and enhanced TJ integrity by promoting the growth of *Lactobacillus* and *Bifidobacterium* spp. [160]. A potential cause underlying this effect is the control of intestinotrophic hormone glucagon-like peptide 2 production, a key element in the regulation of intestinal barrier secreted by endocrine L cells [160]. Furthermore, in rats, fructans stimulated mucosa-associated bifidobacteria which was associated with increased mucus layer and improved mucosal architecture. Consequently, the increase in villus height and crypt depth, in addition to alterations in mucin composition resulted in gut mucosal barrier stabilization [161].

Additionally, SCFA can also contribute to the amelioration of the intestinal barrier. Butyrate is the preferential source of energy for colonic epithelial cells and the most potent acid among the SCFA [162]. Improvement of intestinal epithelial barrier by butyrate likely relies on the expression of TJ proteins [163]. Butyrate can accelerate the assembly of TJ by reorganizing TJ molecules, such as, ZO-1 and occludin, an effect mediated by the activation of AMP-activated protein kinase (AMPK) [163]. In addition, SCFA enhance oxygen consumption by IECs resulting in a reduction in oxygen tension, leading to the stabilization of hypoxia-inducible factor (HIF). Indeed, butyrate increased barrier function and attenuated the infection of *C. difficile* infected mice through the stabilization of HIF-1 [164]. SCFA can also exert intestinal protective mechanisms to the host by altering mucus production and secretion [165]. Mucins are colonic mucous glycoproteins that promote a protective effect against toxic agents through the formation of a mucus layer that acts as physical barrier for the host. SCFA and especially, propionate and butyrate, can reinforce the mucus layer by stimulating mucin2 (MUC2) gene expression, which is the most prominent mucin on the intestinal mucosa surface [166]. The mechanisms that enable butyrate to be involved in MUC2 regulation are mediated via an active region (AP-1) within the MUC2 promotor and histone modifications [166].

In addition to the microbiota-dependent mechanisms for the improvement of the intestinal barrier, several studies have described the direct effects of NDOs on intestinal barrier function. 2′-fucosyllactose and lacto-N-neotetraose promoted enhanced barrier function by increasing the transepithelial resistance in Caco-2Bbe cells [167]. GOS facilitated the tight junction assembly and stabilized the expression and the cellular distribution of the tight junction protein, claudin-3 [155]. Additionally, GOS enhanced mucosal barrier function via the direct stimulation of goblet cells through the up-regulation of gene expression levels of secretory products and Golgi-sulfotransferases in a goblet cell line [168]. Different NDOs have a protective role on intestinal barrier function by differentially affecting epithelial tight junction proteins and goblet cell function, which has been reviewed previously [156].

##### Acidic Environment

SCFA production can lower the pH of the intraluminal space, creating an acidic environment that favors the growth of bifidogenic bacteria. In animal studies, co-administration of GOS with *Bifidobacterium breve* increased the anti-infectious activity against methicillin-resistant *S. aureus* (MRSA), due to a high acetic acid production [112]. Additionally, low intestinal pH and high concentration of acetic acid inhibited Stx production in STEC-infected mice suggesting that such conditions create an unfavorable environment for bacterial pathogens [169]. *E. coli* was also unable to survive inside the acidic environment in FOS- and XOS-fermented cultures leading to growth inhibition [111]. Therefore, the more acidic environment created by SCFA constitutes an unfavorable space for pathogenic bacteria and subsequently inhibits their colonization.

### 4.2. The Effects of NDOs and SCFA against Bacterial Enterotoxins

The cytotoxicity of enterotoxins has been shown to be highly attenuated by the activity of NDOs and SCFA. A summary of their protective effects (both direct and indirect) against bacterial enterotoxins related to human diseases is provided in Table 4. Concerning NDOs, the majority of the mechanisms underlying their anti-virulent behavior rely on receptor mimicry mechanisms and their interference in endocytic pathways, such as the activation of the AMPK protein or the reduction in rRNA depurination. SCFA can also function against enterotoxins through several mechanisms, including metabolic integration, microbiota regulation, inhibition of fluid secretion, and maintenance of intestinal epithelial integrity. The absorption of SCFA in colonic epithelial cells, enable them to influence both extracellular and intracellular host compartments that might subsequently modulate the pathways activated by enterotoxins. However, to date, not all enterotoxins are shown to be inhibited by NDOs and SCFA.

#### 4.2.1. *B. cereus* Enterotoxins

Characterized by their ability to form pores in epithelial cells, leading to fluid release and necrosis, *B. cereus* enterotoxins Hbl, Nhe and CytK do not necessarily need cellular receptors to express their pathogenicity. NDOs with high MW are incapable of crossing the cellular membrane, therefore their role is limited to the extracellular domains of host cells. This further justifies why the majority of therapeutic mechanisms of high MW NDOs focus on outer membrane cellular receptors. To date, no oligosaccharide treatment has been found against *B. cereus-*related enterotoxins, which is most probably due to the lack of cellular receptors and subsequently to the lack of receptor mimicry mechanism. Therapeutic activity against *B. cereus* enterotoxins has not been found either by SCFA. However, since SCFA have been proven to maintain fluid secretion [170], they can probably enhance the extensive influx of Ca^2+^, Na^+^ and efflux of K^+^ caused by *B. cereus* enterotoxins.

#### 4.2.2. Cholera Toxin (CT)

##### NDOs against CT

Several mechanisms contribute to the anti-virulence activity of NDOs against CT. First, COS can suppress intestinal fluid secretion, a key consequence underlying secretory diarrhea induced by CT [171]. Indeed, luminal exposure to COS has been found to reduce intestinal fluid secretion in a mouse model by 30%, a reduction that relies on the activation of AMPK. AMPK is a heterotrimeric protein that apart from its role as a cellular energy conserver, can also mediate epithelial functions, such as tight junction assembly and ion transport (CFTR Cl^−^ channel). COS of low MW (5000 Da) were found to interact with calcium-sensing receptor (CaSR), a G_q_-coupled receptor linked to phospholipase C (PLC) located in IEC. Consequently, through a CaSR-PLC-IP_3_-receptor channel-dependent pathway, COS induce Ca^2+^ secretion from ER and mitochondria, resulting in AMPK activation [171]. Therefore, activated AMPK reduces CT-induced intestinal hypersecretion of chloride, highlighting the substantial anti-diarrheal activity of COS. Since overstimulation of fluid secretion is the consequence of multiple enterotoxins, COS might also be a potential therapy fighting against the adverse effects of other enterotoxins.

The second mechanism arises from direct interaction of NDOs with CT by competing with the GM1 receptor. GM1, the native receptor of CT, is a glycolipid receptor containing a sialylated carbohydrate structure. It is well known that galactose and N-acetylneuraminic acid have a substantial role in the majority of interactions between the receptor and the toxin, since removal of one of these residues confers loss of binding. Such components are also found in the structure of sialylated-oligosaccharides (SOS). In comparison with single monosaccharides, such as lactose, galactose and sialic acid that have been found to be ineffective inhibitors of CT-GM1 binding, the biantennary nature of the glycan chains in SOS showed increased potency for CT-GM1 inhibition [172]. Additionally, 3′-sialyllactose, a predominant sialylated substance in human milk, which partially has the same sequence of the carbohydrate portion of GM1, was also found to behave as a receptor analogue for CT [173]. In addition to acidic HMOs, GOS also contain saccharide residues (e.g., galactosyl residues) that pre-sent similarities with the GM1 structure and therefore showed inhibitory activity against CT binding to the GM1 receptor [174].

Rather than GM1, which is widely the sole receptor for CT intoxication, fucosylated glycan epitopes on glycoproteins were also found to facilitate cell surface binding and endocytic uptake of CT [175]. Although, interaction of the CT binding subunit (CTB) with fucosylated glycans has a much lower affinity than the CTB-GM1 interaction, CTB binding studies demonstrated that low-affinity ligands can be recognized by CTB even in the presence of a much higher affinity ligand. Therefore, based on the functional significance of fucose recognition by CTB, fucosylated molecules can competitively interfere with CTB binding to intestinal epithelial cell lines and primary cells, to prevent CT uptake. Indeed, 2′-FL, a fraction identified also in HMOs mixtures, was shown to inhibit CTB binding to GM1 [176]. The appearance of additional binding sites is further justified by a study in which 20 of the most abundant HMOs were tested against CT and despite their affinity, the binding site was found to be distinct from the one of the native receptor binding sites on CT [177]. Dendrimers, as obtained by synthetic conjugation of GM1-oligosaccharides, yielded very potent inhibitors of CTB with picomolar potencies [178] as evaluated by binding studies using intestinal organoids [179]. Additionally, simplified polymeric ligands were shown to be very potent, even when incorporating fucose derivatives [180,181].

##### SCFA against CT

Cytotoxic consequences of CT such as watery diarrhea, are directly linked with low fluid absorption and hypersecretion of electrolytes and water in the intestinal lumen. SCFA can enhance the impairment of colonic functions occurring during CT pathogenicity by stimulating colonic absorption and reducing net fluid loss. Even if fluid secretion stimulated by CT mostly occurs from the small intestine, the colon can also bind with CT and secrete fluid and electrolytes after exposure to purified CT. An in vivo study showed that SCFA (acetate, propionate, butyrate) can significantly reduce the secretion of water and electrolytes (Na^+^, K^+^, Cl^−^) in the colon of a CT-induced rabbit model. The anti-secretory behavior of SCFA is likely a result of their pro-absorptive effects on Na^+^ and Cl^−^ transport. Interestingly, similar inhibitory effects of SCFA were not observed in the case of HCO_3_^−^ secretion [182].

#### 4.2.3. *C. difficile* Enterotoxins

##### NDOs against *C. difficile* Toxin A (TcdA) and Toxin B (TcdB)

Two substantial oligosaccharides, HMOs and FOS, show anti-virulent activity against *C. difficile* large toxins, TcdA and TcdB. Both enterotoxins incorporate two distinct functional regions in their structure, i.e., the CROPs and the region N-terminally adjacent to the CROPs, which independently serve as receptor-binding domains. Multiple receptors can bind TcdA and TcdB whereas glycans were identified as high-affinity binding structures for TcdA and specific protein receptors were identified for TcdB. However, both of them have carbohydrate-binding sites that bind to HMOs [183]. Concerning TcdA, a variety of glycans, including the linear B type 2 trisaccharide α-Gal-(1,3)-β-Gal-(1,4)-β-GlcNAc, can bind either to TcdA or to a part of the TcdA CROPs [184]. Therefore, HMOs that present structural similarities with the cellular receptors, can inhibit toxin binding. Such structures were found to be LNFPV and LNnH, two HMO structures, which demonstrated high binding affinity to TcdA, thereby obstructing toxin binding to its native receptor [185]. Interestingly, molecular docking analysis of the two aforementioned compounds showed stronger binding to the TcdA binding site than that of α-Gal-(1,3)-β-Gal-(1,4)-β-GlcNAc, which further potentiates the role of HMOs in in-hibiting toxin binding [185].

Additionally, FOS have protective effects against *C. difficile* infections by inhibiting the expression of toxin-related genes. FOS decreased the gene copy numbers of the *Clostridium* cluster XI and of the *C. difficil*e toxin B (TcdB) in the fecal microbiota of rats in an inflammatory bowel disease model, correlating with the reduction in chronic intestinal inflammation [186] and nosocomial diarrhea. The ability of FOS to exert anti-inflammatory effects possibly relies on the formation of SCFA, although further research is needed.

##### SCFA against *C. difficile* Toxin A (TcdA) and Toxin B (TcdB)

Multiple SCFA have been shown to efficiently reduce cytotoxicity of *C. difficile* enterotoxins, predominately via indirect pathways. Anti-pathogenic mechanisms of increased SCFA concentrations mostly rely on their ability to create an acidic luminal environment. Using in vitro experiments, the relationship between *C. difficile* enterotoxins’ production with different concentrations of SCFA (acetate, propionate, butyrate) and pH levels was investigated. Results pointed out inhibitory effects of SCFA on the growth and the production of *C. difficile* enterotoxins, while beneficial effects were related to elevated SCFA concentration and lower pH levels [187]. In addition, SCFA, and especially butyrate, exhibits protective effects against CDI by restoring the damage of IECs induced by *C. difficile* enterotoxins. A recent study was conducted to examine the mode of action of butyrate against CDI and revealed that even if no effects on bacterial colonization or *C. difficile* enterotoxin production was observed, butyrate managed to attenuate intestinal inflammation and improved the intestinal barrier function in CD-infected mice by acting directly on IECs. The reduction in intestinal epithelial permeability induced by butyrate was achieved via an HIF-1a-dependent mechanism. Administration of butyrate in mice infected with *C. difficile* demonstrated elevated levels of *Hif1a* expression and stability that is a relevant effect for intestinal barrier integrity. Through the stabilization of HIF-1, damage of IECs caused by *C. difficile* toxins, was repaired thereby preventing the local inflammatory response and systemic implications [164]. In comparison with the large clostridial enterotoxins (TcdA, TcdB), so far, no effect has been reported by NDOs and SCFA against the CDT enterotoxin produced by *C. difficile*.

#### 4.2.4. *C. perfringens* Enterotoxins

The binding of the two human related *C. perfringens* enterotoxins, CPE and CPB, to intestinal epithelial cells relies on the presence of tight and gap junctions. CPE binds strongly to claudin-3, -4 cellular receptors [188], while CPB binds to pannexin receptor (P2X7) that belongs to gap junction proteins, proteins that are responsible for intracellular communication [21]. Dietary components, such as NDOs and their metabolites are known to regulate intestinal barrier function by changing the expression and the distribution of junction proteins [155,189]. However, no interaction between these specific receptors and NDOs and SCFA has been identified so far.

On the other hand, SCFA have shown potential against CPE by inhibiting the spore formation stage. Unlike other enterotoxins that are produced and released during the replicative cycle of bacteria, CPE is induced during sporulation. SCFA can inhibit spore formation and subsequently enterotoxin production, thereby preventing associated undesirable consequences, such as antibiotic-associated diarrhea. Indeed, a study has shown that four SCFA (acetate, isobutyrate, isovalerate, succinate) produced by *Bacteroides fragilis* can lower the amount of heat-resistant spores or even reduce the number of viable cells as observed for isobutyrate [190]. Therefore, fermentation products of *Bacteroides* spp. can inhibit sporulation of *C*. *perfringens* and thus prevent CPE production.

#### 4.2.5. Heat-Labile (LT) and Heat-Stable (ST) Enterotoxins

##### NDOs against LT and ST Enterotoxins

NDOs have shown anti-pathogenic functionalities against LT and ST enterotoxins, which are both produced by virulent *E. coli* strains. The protective role of HMOs and specifically of fucosylated oligosaccharides against ST enterotoxins has been identified by using a suckling mouse model [191]. Additional studies that examine structurally similar HMOs proved similar ef-fects against STa in T84 intestinal cells [192]. The mechanism that potentially underlies this effect relies on the allosteric binding of HMOs to the STa receptor. Even if predominately NDOs bind directly to enterotoxins in order to inhibit their action, in the case of STa, fucosylated oligosaccharides bind preferentially to the STa receptor. Cytotoxicity of STa begins with the binding to GC-C that subsequently leads to the activation of the GC-C intracellular catalytic domain. Fucosylated fractions of human milk were found to block activation of human GC by binding allosterically to GC-C and therefore prevent STa infection [192].

HMOs can also bind directly to LT although at a distinct position of the native LT receptor (GM1). However, an evidence of competitive binding between GM1 and two HMOs (2′-fucosyllactose and lacto-N-fucopentaose I) gives the perspective of a receptor mimicry mechanism [177]. Besides the direct effect of HMOs on LT, FOS can also interfere with the cytotoxicity of LT. By acting synergistically with *Lactobacillus rhamnosus*, FOS through an in vitro study were found to significantly inactivate ETEC by decreasing LT production [193]. The reduction in the amount of enterotoxin is possibly an indirect effect of FOS that derives from the production of SCFA metabolites.

##### SCFA against LT Enterotoxin

SCFA can disturb the production of LT, however no mechanism has been determined so far. A study investigating the effect of SCFA on the production of LT enterotoxin showed that the addition of SCFA with different carbon chain length can significantly reduce or abolish LT production. In addition to the three main products of oligosaccharide fermentation, acetic (C-2), propionic (C-3) and butyric acid (C-4), three more SCFA presented inhibitory effects against LT, including *n*-valeric (C-5), *n*-caproic (C-6) and *n*-heptylic acids (C-7). The effectiveness of SCFA at a concentration of 2 mg/mL was proportional to the elongation of carbon chain length from C-2 to C-7. Eventually, *n*-heptylic acid (C-7) showed the most intense inhibition of LT production, while longer chain fatty acids tested (C-8 to C-10), inversely recovered the LT levels [194]. Since production of LT is essential for the induction of diarrhea, by reducing the LT production, diarrheal consequences can be prevented.

#### 4.2.6. Shiga Toxins (Stxs)

##### NDOs against Stxs

Specific NDOs such as POS and HMOs, have the capability to reduce Stx cytotoxicity. POS derived from the hydrolysis of citrus and apple pectin were found to completely protect human colonic HT29 cells from the toxic effects of Stx1 and Stx2 [195]. The structure of POS is mostly composed of a GalA-rich backbone, a carbohydrate moiety similar, but distinct from the structure of the Gb3 receptor, thereby receptor mimicry may not be involved [140]. However, POS can minimize Stx cytotoxicity by reducing rRNA depurination of host cells, an effect caused by the enzymatically active Stx A1 fragment that subsequently leads to protein synthesis inhibition [196]. Indeed, reduction in rRNA depurination induced by POS was proved through a study that was based on a TaqMan probe-based RT-qPCR analysis [140]. This further suggests that POS might block the entry of Stx into cells.

In addition to the relationship of Stxs with POS, a variety of HMOs have been shown to bind with several enterotoxins, including Stxs [177]. Such potential of binding was further investigated and the measured binding affinities to HMOs were found to be lower in comparison with the affinity of Stxs to the Gb3 analogue, P^k^ trisaccharide. However, apparent association constants K_a,app_ for HMOs binding to Stx, was found to be similar to many biologically relevant carbohydrate–protein interactions, therefore HMOs could compete with such monovalent interactions. Surprisingly, even if HMOs manage to bind efficiently with Stxs, competitive assays proved that the specific binding occurred at a distinct position rather than at the Gb3 binding site [177]. Given these results, receptor binding inhibition seems unlikely to occur, therefore additional inhibitory mechanisms might be involved and further research on the anti-pathogenic effects of HMOs against Stxs should be established.

##### SCFA against Stxs

Several investigations of SCFA have also revealed a wide range of anti-pathogenic functionalities against Stxs. Acetic acid and lactic acid, also produced by beneficial gut bacteria, such as, lactic acid bacteria, have shown to decrease *stx2* gene expression. The reduction in *stx2* gene expression leads to diminished enterotoxin-associated cytotoxicity that is likely a consequence of low pH conditions as a result of organic acid production [169,197]. Additionally, acetate promotes anti-virulence activity against *E. coli* 0157 by inhibiting the translocation of Stx across the colonic epithelial monolayer [198]. A study has shown that the protective role of certain bifidobacteria against pathogenic *E. coli* 0157 relied on high production of acetate. Acetate induces the expression of three specific genes, *Apoe*, *C3* and *Pla2g2a,* that have been found to mediate cellular energy metabolism and anti-inflammatory responses in the colonic epithelium. Moreover, increased production of acetate prevented the reduction in the transepithelial electrical resistance caused by an *E. coli* 0157 infection, thereby inhibiting Stx translocation across the colonic epithelium monolayer from the luminal toward the basolateral side [199]. On the other hand, butyrate acts unfavorably against Stx expression, since butyrate upregulates Gb3 production. In an animal study, increased expression of Gb3 on intestinal and kidney tissue is assumed to be a consequence of a butyrate-dependent mechanism causing detrimental effects on the infected host [200].

#### 4.2.7. Staphylococcal Enterotoxins (SEs)

To date, to the best of our knowledge, there is no evidence of inhibitory effects of NDOs against SEs. The cytotoxicity pathway of SEs starts with their binding to MHC II class molecules after crossing the epithelial barrier. Binding of SEs to MHC II class constitutes a pivotal step for their cytotoxicity and is one of the main therapeutic targets. The inactivity of NDOs against SEs might be related to the difficulty of high MW NDOs to cross the epithelial barrier, leading to a diminished chance for interaction with intracellular targets such as MHC II class. On the other hand, SCFA and especially butyrate is a well-known HDAC inhibitor and thus can downregulate pro-inflammatory mediator expression leading to increased regulatory T cell differentiation [201,202] However, even if SCFA have been proved to modulate host immune responses, no effect of SCFA was reported on MHC class II [203].

## 5. Conclusions and Future Perspectives

This review presents an overview of a great number of NDOs and SCFA, as their fermentation products produced by the gut microbiota, that could be considered as therapeutic agents to limit the cytotoxic effects on the human intestine induced by EPB and their toxins. Enterotoxins can drastically damage intestinal epithelial cells, leading to gastrointestinal and systematic complications. To exert their cytotoxicity, enterotoxins mostly alter cell viability through the inactivation or the cleavage of intracellular targets or by promoting the efflux of water and electrolytes through the formation of membrane pores. A thorough exploration of the key characteristics of intestinal pathogens and their toxins, resulted in the selection of various NDOs and SCFA as potential treatments fighting against EPB-related infections.

NDOs interact with EPB and thereby inhibit their activity following either direct or indirect mechanisms. Direct effects of NDOs on EPB, including anti-adhesive, anti-biofilm and anti-growth effects, partly rely on receptor mimicry strategies. The structural similarity between adhesive receptors of EPB and free oligosaccharides, enable NDOs to mimic bacterial binding to host cell surface, thereby blocking bacterial adherence. The anti-microbial potential of NDOs is not only governed by their structural similarity with the receptors, since other characteristics, such as e.g., their charge also endows them with additional potency to interact with EPB. However, not all related studies proposed a thorough explanation for the direct anti-pathogenic effects of NDOs. Consequently, a more elucidative characterization of NDOs along with a wider range of pathogens, could reveal further details of their anti-microbial capabilities. On the other hand, indirect strategies of NDOs, rely on the maintenance of gut homeostasis, mediated by the promotion of beneficial microbiome and subsequently by the activity of SCFA. Additionally, barrier-protecting and immune-related properties of NDOs and SCFA can also contribute to the maintenance of gut homeostasis.

Except the indirect and direct anti-pathogenic effects of NDOs on EPB, NDOs and SCFA influence the cytotoxicity of EPB-associated enterotoxins. The anti-toxin activity of NDOs mostly derives from the blockage of the enterotoxin adherence, as the primary step of the toxin-induced cytotoxicity pathway. Structural similarity between NDOs and toxin receptors leads to inhibition of toxin adherence. Despite their effects on the extracellular compartments, LMW NDOs have been shown to influence intracellular enterotoxin pathways, leading to the reduction in fluid hypersecretion. Similar anti-toxin potentials are also identified by SCFA, since their low MW enable them to cross the intestinal epithelium barrier and thereby target intracellular targets (e.g., HIF-1 factor). However, the most prominent cause of anti-toxin effects induced by SCFA is the creation of an acidic environment that results in unfavorable conditions for the enterotoxins. To uncover additional targets towards the pathways of enterotoxins, further research is warranted to elucidate adhesive mechanisms and eventually lead to the optimal NDOs or SCFA treatment corresponding to specific enterotoxin mechanisms.

Based on the findings of this review, NDOs (e.g., HMOs, GOS and POS) that acquire the highest structural resemblance with host cells receptors for enterotoxins demonstrated the highest anti-pathogenic capacity. Given the promising anti-pathogenic potential of NDOs, further research related to appropriate utilization of NDOs and SCFA through a clinical approach is required. Although food consumption and production vary between cultures, regions, and countries, NDOs are present in a wide variety of sources found in the regular human diet [204]. Different individual factors, like metabolism, gender, age, genetic variations and microbiome composition will influence the response to these dietary interventions, therefore, it is complicated to suggest a general sufficient amount and source of NDOs. In this regard, personalized nutrition will be more suitable than population-based nutritional advice [205]. NDOs are mostly tested as individual agents, but it could be interesting to evaluate the combination of NDOs with different SCFA to indicate if there is a possibility to exert a synergistic effect. To date, antibiotics are considered as the major therapeutic strategy to address infectious diseases. However, the improper and high use of antibiotics result in a decreased susceptibility and increased resistance against these antimicrobials. The use of NDOs and SCFA in lieu of or in combination with antibiotics to control infectious diseases might contribute to a reduction in this emerging antibiotic resistance. Additionally, several NDOs (such as GOS, FOS) have been already added in infant nutrition formulas in an attempt to mimic the endogenous HMOs. Thereby, NDOs already offer a safer and non-toxic alternative for anti-microbial therapy. In this respect, NDOs and SCFA provide specific, targeted activity and are less likely to present negative side effects in comparison with commensal treatments such as antibiotics. Given the burden and long-term consequences of microbial-associated gastrointestinal diseases, the route of using NDOs and SCFA to fortify gut flora and encounter EPB and their toxins holds much potential. Therefore, we anticipate that with additional focused studies, these anti-toxin molecules could soon reach the optimal goal to be used as therapeutic agents with great impact on the treatment of infectious diseases.

## Figures and Tables

**Figure 1 toxins-13-00175-f001:**
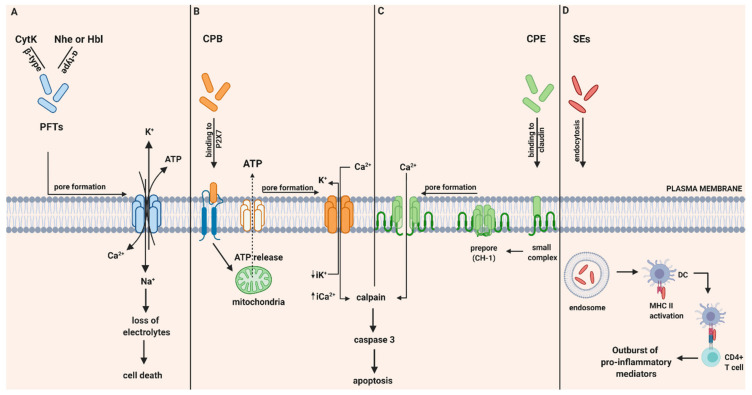
Intoxication pathways of hemolysin B (Hbl), non-hemolytic enterotoxin (Nhe), cytotoxin K (Cyt K), CPB, CPE and staphylococcal enterotoxins (SEs). (**A**) Proposed mechanism of *B. cereus* enterotoxins, Hbl, Nhe and Cyt K. Hbl, Nhe and CytK begin their cytotoxic pathway by forming cell membrane pores that lead to the influx of Ca^2+^ and Na^+^ and the efflux of K^+^ and ATP and thereby to the loss of electrolytes and cell death. (**B**) Intracellular action of *C. perfringens* beta-toxin (CPB). Once CPB binds to ATP-gated P2X7 receptor, ATP is released from target cells to ATP-release channel pannexin 1. Induced by ATP release, CPB is oligomerized and a pore is formed. Pore formation leads to increased influx of Ca^2+^ that triggers calpain activation and necroptosis. Furthermore, pore formation also results in loss of intracytoplasmic K^+^ (iK^+^) that is associated with the activation of MAPK and JNK that are responsible for host cell survival and defense pathways. (**C**) Intracellular action of *C. perfringens* enterotoxin (CPE). CPE binds to its cellular receptor, claudin, and forms a small complex. Later on, six small CPE complexes oligomerized forming a prepore on the plasma membrane called CH-1. Assembly of β-hairpin loops into a β-barrel structure allows a cation-permeating pore insertion in the plasma membrane. The influx of Ca^2+^ stimulates calpain activity and thus the activation of caspase-3 and apoptosis. (**D**) SE-associated gastrointestinal (GI) inflammatory injury. Once they get endocytosed, SEs bind to MHC II class molecules and subsequently attract CD4^+^ T cells. Afterwards, an excessive production of pro inflammatory chemokines and cytokines is induced. Created with biorender.com (accessed on 28 May 2020).

**Figure 2 toxins-13-00175-f002:**
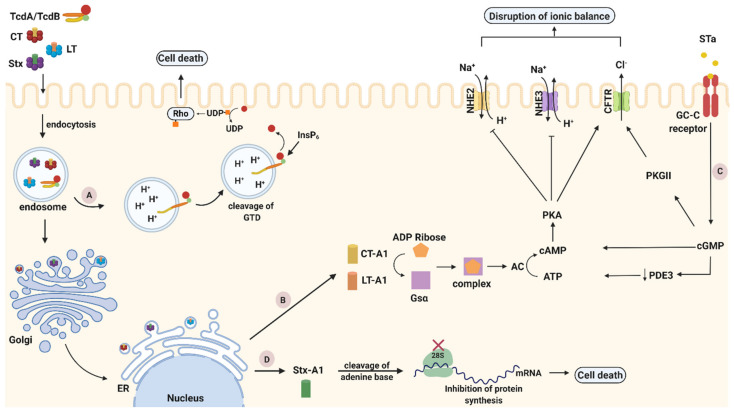
Intoxication pathways of cholera toxin (CT), heat-labile toxin (LT), Shiga toxin (Stx) and *C. difficile* toxins A and B (TcdA/TcdB). (**A**). TcdA/TcdB cytotoxicity pathways. After internalization through endocytosis, toxins reach acidified endosomes. Low pH induces structural conformations in the toxin delivery domain leading to pore formation and translocation of glucosyltransferase domain (GTD) into the host cytosol. Rho family proteins become inactivated when GTD transfers a glucose unit from uridine diphosphate (UDP)-glucose to the switch I region of GTPase, leading to pathogenic effects or cell death. (**B**) CT and LT cytotoxicity pathways. After endocytosis and travelling of CT and LT as holotoxins through the trans Golgi network (TGN) and the ER, their catalytic A1 subunit is cleaved and released inside the cytoplasm. Thereafter, the A1 fragment ADP-ribosylate the Gsα subunit of G-protein and consequently activates adenylyl cyclase (AC). Activation of AC leads to elevated levels of cyclic AMP (cAMP) that activate protein kinase A (PKA). PKA stimulates the secretion of Cl^−^ through cystic fibrosis transmembrane regulator (CFTR) but provokes the inhibition of Na^+^ absorption leading to the disturbance of cellular ionic balance, and ultimately apoptosis. (**C**) STa cytotoxicity pathway. STa binds to guanylate cyclase C (GC-C) receptor and activates its intracellular catalytic domain, which induces cyclic GMP (cGMP). Elevated levels of cGMP activate cAMP-dependent PKA and inhibit phosphodiesterase 3 (PDE3). Subsequently, activated cAMP-dependent PKA along with PKGII, phosphorylate and open cystic fibrosis transmembrane conductance regulator (CFTR) Cl^−^ channel. Through CFTR, Cl^−^ and HCO_3_^−^ are released in the intestinal lumen, while Na^+^ reabsorption is inhibited as PKA has the ability to block the NHE3 channel. These modulating effects on the ionic channels induced by STa, finally result in an electrolyte imbalance that causes cell death. (**D**) Stx cytotoxicity pathway. Stx is internalized inside host cells (endocytosis) within early endosomes. Afterwards, Stx is following a retrograde transportation pathway, which is directed towards the transGolgi network (TGN) and subsequently reaches the endoplasmic reticulum (ER). In the ER, the enzymatically active A1 fragment translocates into the cytoplasm. Thereafter, it cleaves one adenine residue from the 28S RNA of the ribosomal subunit and thus inhibits protein synthesis leading to cell death. Created with biorender.com (accessed on 20 November 2020).

**Table 1 toxins-13-00175-t001:** Structural overview of different non-digestible oligosaccharides (NDOs).

Non-Digestible Oligosaccharides	Structures
Alginate-oligosaccharides (AOS)	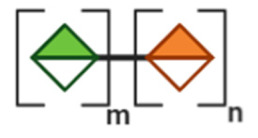
Chito-oligosaccharides/Chitosan-oligosaccharides (COS)	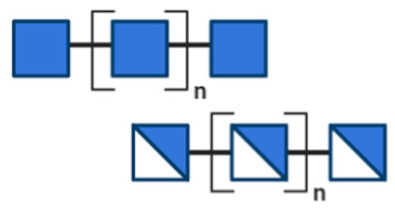
Fructo-oligosaccharides (FOS)	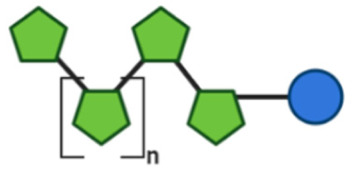
Galacto-oligosaccharides (GOS)	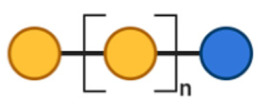
Human milk oligosaccharides (HMOs)	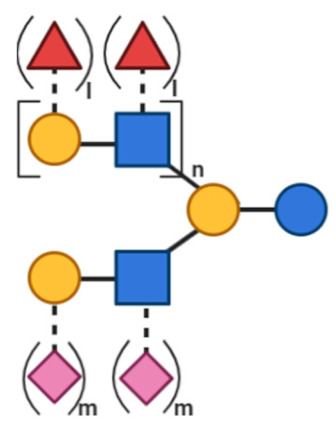
Isomalto-oligosaccharides (IMOS)	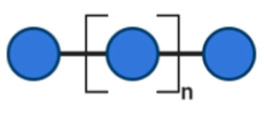
Mannan-oligosaccharides (MOS)	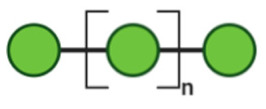
Pectic-oligosaccharides (POS)	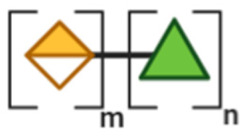
Xylo-oligosaccharides (XOS)	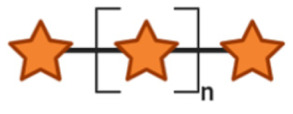

Key monosaccharides: fructose (Fru,
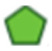
); fucose (Fuc,
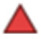
); galactose(Gal,
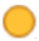
); galacturonic acid (GalA,
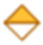
); glucosamine (GlcN,
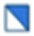
); N-acetylglucosamine (GlcNAc,
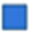
); glucose (Glc,
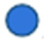
); guluronic acid (GulA,
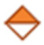
); mannose (Man,
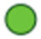
); mannuronic acid (ManA,
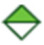
); rhamnose (Rha,
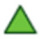
); sialic acid (Sia,
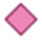
); xylose (Xyl,
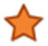
).

**Table 2 toxins-13-00175-t002:** Indirect effects of NDOs on enterotoxin-producing bacteria.

NDOs	Beneficial Bacteria	Bacteria Producing Enterotoxins	Mechanism of Action	Health Benefit	Ref.
FOS	*Lactobacillus rhamnosus* *Lactobacillus reuteri* *Lactobacillus rhamnosus* (probiotic mixture) mixture of six *Lactobacilli* and three Bifidobacteria strains	Enteropathogenic *Escherichia coli* (EPEC), *Clostridium difficile*	Not investigated	Growth inhibition	[108]
FOS, inulin	*Lactobacillus rhamnosus*	Enterotoxigenic *Escherichia coli* (ETEC)	Decrease in cAMP and cGMP levels	Anti-virulence activity	[109]
FOS, inulin	*Lactobcillus acidophilus*, *Lactobacillus lactis*, *Lactobacillus casei*, *Lactobacillus reuteri*	*Staphylococcus aureus*, *Escherichia coli*	Production of metabolites by beneficial bacteria	Anti-biofilm, anti-adhesive, antimicrobial effect	[110]
FOS, XOS, inulin, mixtures of inulin:FOS (80:20 *w*/*w*) and FOS:XOS (50:50 *w*/*w*)	*Lactobacillus plantarum*, *Bifidobacterium bifidum*	*Escherichia coli*	Production of SCFA	Growth inhibition of *Escherichia coli*	[111]
XOS	*Lactobacillus* spp. *Bifidobacterium* spp.	*Clostridium perfringens*	Production of SCFA	Growth suppression of *Clostridium Perfringens*	[103]
GOS	*Bifidobacterium breve*	Methicillin-resistant *Staphylococcus aureus* (MRSA)	High production of acetic acid (low pH)	Prevention of systemic MRSA infection	[112]
GOS, IMOS	*Lactobacillus plantarum*, *Lactobacillus paracasei*, *Bifidobacterium breve*, *Bifidobacterium lactis*	*Clostridium difficile*	Stimulation of gut microflora	Protective effect against CDI	[113]

**Table 3 toxins-13-00175-t003:** Direct effects of NDOs on enterotoxin-producing bacteria.

Bacteria	Toxins	NDOs	Mechanism of Action	Ref.
*Escherichia coli*	Stx, LTs, STs	Pooled HMOs, 3′SL, 6′SL, DSLNT, 6SLN, 3′S3FL, LST a	Inhibition of hemagglutination	[114]
		Fucosylated HMOs	Anti-adhesive effect	[115]
		Pooled HMOs, 3-FL, 3′SL, 6′SL, acidic and neutral (HMW and LMW) HMOs	Anti-adhesive effect	[116]
		2′FL, 3-FL	Anti-adhesive effect	[117]
		2′FL, 6′SL	Anti-adhesive effect	[118]
		Pooled HMOs	Reduced enteropathogenic *E. coli* (EPEC) attachment to epithelial cells	[119]
		AOS	Inhibition of biofilm formation	[120]
		COS	Antimicrobial effect	[121]
			Anti-adhesive effect	[122]
			Antibacterial effect	[123,124]
			(growth inhibition)	
			Antioxidant and antimicrobial effect	[125]
			Membrane disruption (growth inhibition)	[126]
		GOS	Anti-adhesive effect	[127]
		MOS	Anti-adhesive effect	[128,129]
		POS	Growth inhibition	[130]
			Anti-adhesive effect	[131]
			Antimicrobial effect	[132,133]
*Vibrio cholerae*	CT	Pooled HMOs	Anti-adhesive effect (neutral, HMW)	[116]
		Isolated HMOs	No anti-adhesive effect (LMW)	[116]
		COS, NAc-COS	Bactericidal effect	[134]
*Bacillus cereus*	Hbl, Nhe, CytK	COS	Antibacterial effect	[134,135]
			(growth inhibition)	
			Membrane disruption (growth inhibition)	[126]
*Staphylococcus aureus*	SEs	Pooled HMOs	Inhibition of biofilm formation	[105]
		COS	Antibacterial effect	[123,134,136]
			(growth inhibition)	
			Antioxidant and antimicrobial effect	[125]
		POS	Growth inhibition	[130]
			Antimicrobial effect	[132]
			Anti-adhesive effect	[137]
		XOS	Antibacterial effect	[138]
*Clostridium* spp.	TcdA, TcdB, CDT, CPE, CPB	FOS	Anti-adhesive effect Inhibition of biofilm formation	[139]

**Table 4 toxins-13-00175-t004:** Effects of NDOs and short chain fatty acids (SCFA) against bacterial enterotoxins.

Enterotoxins	NDOs/SCFA	Mechanism of Action	Health Benefit	Ref.
Cholera toxin	COS	Activation of AMPK	Reduction in CT-induced intestinal fluid secretion	[171]
Cholera toxin	SOS	Mimicking GM1 receptor	Anti-adhesive effect	[172]
Cholera toxin	Pooled HMOs	GM1 mimicking	Inhibition of sialyllactose on fluid accumulation induced by CT	[173]
Cholera toxin	GOS	Mimicking GM1 receptor	Anti-adhesive effect	[174]
Cholera toxin	2′-FL	2′-FL resembles fucosylated glycan epitopes	Inhibition of CTB binding	[176]
Cholera Toxin	2′FL, LNnT, LNFP I, II, III	Binding with CTB	No inhibitory effect	[177]
Cholera toxin	Acetate, propionate, butyrate	Reduction in water and electrolyte secretion	Anti-diarrheal effects	[182]
Shiga toxins	2′FL, LNnT, LNFP I, II, III	Binding with STX	No inhibitory effect	[177]
Shiga toxins	POS	POS mimic the interaction with the galabiose receptor	Inhibition of Stx	[195]
Shiga toxins	POS	Reduction in rRNA depurination	Reduction in Stx cytotoxicity	[140]
Shiga toxins	Acetic acid, Lactic acid	Decrease in pH	Reduction in stx2 gene expression	[169,197]
Shiga Toxins	Acetate	Reduction in transepithelial electrical resistance	Prevention of Stx translocation	[198,199]
Heat-stable enterotoxin (ST)	Fucosylated HMOs	not determined	Protective effect (suckling mice)	[191]
Heat-stable enterotoxin (ST)	Pooled HMOs, Fucosylated HMOs	Block of human guanylate cyclase activation by allosteric binding to the STa receptor	Protective effect (T84 cells)	[192]
Heat-labile enterotoxin (LT)	FOS	Bifidogenic effect with *L. rhamnosus*	Reduction in growth and toxin production	[193]
Heat-labile enterotoxin type 1 (LT-1)	2′FL, LNnT, LNFP I, II, III	Binding with LTB	No inhibitory effect	[177]
Heat-labile enterotoxin (LT)	Acetic, propionic, butyric, valeric, caproic and heptylic acids	Effects on the biosynthesis of LT	Reduction in LT production	[194]
*C. difficile* toxin B (TcdB)	FOS, inulin	Inhibiting toxin-related gene expression	Anti-inflammatory effect	[186]
*C. difficile* toxins A (TcdA) and B (TcdB)	Pooled HMOs	Mimicking the structure of cellular receptors	Inhibition of toxins’ binding to cellular receptors	[185]
*C. difficile* toxins A (TcdA) and B (TcdB)	Acetate, propionate, butyrate	Decrease in pH	Prevention of growth and elaboration of toxin (colonization resistance)	[187]
*C. difficile* toxins A (TcdA) and B (TcdB)	Butyrate	Stabilization of HIF-1	Anti-inflammatory effects and intestinal barrier function improvement	[164]
*C. perfringens* enterotoxin	Acetate, isobutyrate, isovalerate, succinate	Inhibition of toxin sporulation	Reduction in enterotoxin cytotoxicity	[190]

## Data Availability

Not applicable.

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
