# Peer review of "Non-Digestible Oligosaccharides and Short Chain Fatty Acids as Therapeutic Targets against Enterotoxin-Producing Bacteria and Their Toxins"

_toxins, 2021, doi:10.3390/toxins13030175_

Round 1
Reviewer 1 Report
I would like to thank the Authors, it is one of the best review that I ever read.
There are present some little mistakes that I assume are due to bad formatting:
Line 106: B. cereus in italic
Line 196: C. perfringens in italic
Line 249: S. aureus in italic
Line 332: C. difficile in italic
Line 376: C. difficile in italic
Line 419: V. cholerae in italic
Line 496: E. coli in italic
Line 579: E. coli in italic
Line 1010: B cereus in italic
Line 1080-81: C.difficile in italic
Line 1104: C.difficile in italic
Line 1126: C. perfringens in italic
Reviewer 2 Report
This review is excellently written and interesting for the readers of "Toxins".
I have a few (very little) suggestions:
Page1, line 34: is the comma behind (SCFA) correct?
Page 5, line 216: I would shortly name Claudins as receptor here, or alternatively write (see below) to make clear that receptors are known.
Figure 1B: looks like ATP could diffuse through the membrane. Although written in the text, I would add the Pannexins to the figure.
Page 8, line 343: should be an ABCD model.
Page 11, line 484: bacillus
Page 13, line 599: an AB5 protein
Figure 2: In A, the toxin TcdA/TcdB is in the cytosol. This is misleading. Simply remove the toxin symbol here.
Table 1, legend: At least in my printed version of the review, the symbols are not at correct positions, e.g. the yellow circle hides the word Table.
Table 2: Staph aureus: Is "No effect" correct? I do not understand its meaning here.
Page 22, line 916: do you mean Isobutyric acid?
Page 23, line 973: what do you mean with detection of AP1?
Page 23, line 985: space missing before "Acidic environment"
Table 3 (headings): The En- from Enterotoxins may be removed to the line below
Page 24, line 997: only here, you used an s behind SCFA.
Page 25, line 1020: should be have been proven.
Page 25, line 1028: I think it should be apart from?
Page 16, line 1086: For me TcdB,S makes no sense, did you mean TcdB?
It would be interesting to add information about the amount of NDO and SCFA necessary for benefit and whether the level of normal food intake (e.g. fruits, an apple a day...;)) would be sufficient to elevate one of the effects described.
Could you add a short comment on this?
I read the review with great interest, thank you for this work.
